# Exploration of novel αβ-protein folds through de novo design

Shintaro Minami [1,8], Naohiro Kobayashi[2,3,8], Toshihiko Sugiki [2], Toshio Nagashima[3], Toshimichi Fujiwara [2], Rie Tatsumi-Koga[1], George Chikenji [4] & Nobuyasu Koga [1,5,6,7] ✉

A fundamental question in protein evolution is whether nature has exhaustively sampled nearly all possible protein folds throughout evolution, or whether a large fraction of the possible folds remains unexplored. To address this question, we defined a set of rules for β-sheet topology to predict novel αβ-folds and carried out a systematic de novo protein design exploration of the novel αβ-folds predicted by the rules. The designs for all eight of the predicted novel αβ-folds with a four-stranded β-sheet, including a knot-forming one, folded into structures close to the design models. Further, the rules predicted more than 10,000 novel αβ-folds with five- to eight-stranded β-sheets; this number far exceeds the number of αβ-folds observed in nature so far. This result suggests that a vast number of αβ-folds are possible, but have not emerged or have become extinct due to evolutionary bias.

The structural diversity of proteins underlies their functional variety. The overall structure of a protein is determined by its fold, that is, the spatial arrangement of, and connections between, the secondary structure elements. Hundreds of thousands of naturally occurring protein structures have been solved and deposited in the Protein Data Bank (PDB), and the number continues to grow. However, in recent years, novel protein folds have rarely been discovered[1–3], suggesting that nearly all folds existing in nature have been found. This does not necessarily indicate that all folds accessible to the polypeptide chain have been uncovered. Although debated[4–7], it has been suggested that nature may have sampled only a small fraction of the possible fold space during evolution[5–7]. We investigated this hypothesis through de novo protein design for the folds that have not been sampled by natural evolution.

Recently developed principles for designing protein structures have made possible the design of a wide range of new proteins from scratch[8–12], allowing exploration of the huge sequence space beyond that sampled by natural evolution. However, exploration of the fold space has so far been limited to naturally occurring protein folds[8–12], except for one new fold of a protein called Top7 (ref. 13). To explore the fold space beyond that sampled by natural evolution, a 'map' to search for the folds that are possible, but not observed in nature (that is, novel folds), is indispensable. Therefore, we defined a set of rules for β-sheet topology to predict novel αβ-folds, and we carried out a systematic exploration of novel αβ-folds through de novo protein design, guided by these rules.

## Results

### αβ-Folds not observed in nature

The αβ-folds, most of which are involved in enzymatic functions[14], account for more than half of the protein folds identified so far[15]. We first sought to identify unobserved αβ-folds with a three- to eight-stranded open β-sheet, that is, a β-sheet that does not form a barrel. We defined αβ-folds in a more abstract manner on the basis of their β-sheet topology, that is, the number, order and orientation of constituent β-strands in a β-sheet (Fig. 1a). Moreover, we considered

[1]Protein Design Group, Exploratory Research Center on Life and Living Systems (ExCELLS), National Institutes of Natural Sciences (NINS), Okazaki, Japan. [2]Institute for Protein Research (IPR), Osaka University, Osaka, Japan. [3]RIKEN Center for Biosystems Dynamics Research, RIKEN, Yokohama, Japan. [4]Department of Applied Physics, Graduate School of Engineering, Nagoya University, Nagoya, Japan. [5]SOKENDAI, The Graduate University for Advanced Studies, Hayama, Japan. [6]Research Center of Integrative Molecular Systems, Institute for Molecular Science (IMS), National Institutes of Natural Sciences (NINS), Okazaki, Japan. [7]Present address: Laboratory for Protein Design, Institute for Protein Research (IPR), Osaka University, Osaka, Japan. [8]These authors contributed equally: Shintaro Minami, Naohiro Kobayashi. ✉e-mail: nkoga@protein.osaka-u.ac.jp

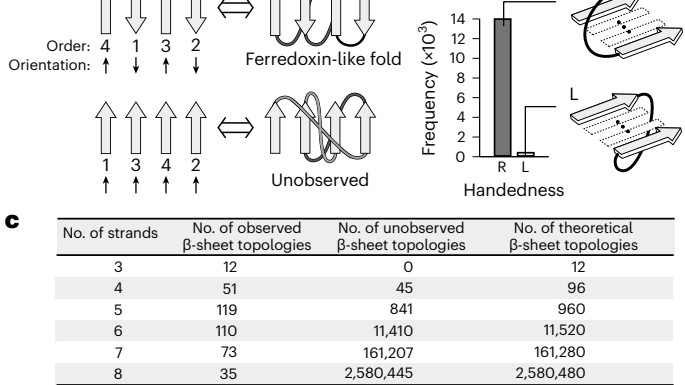

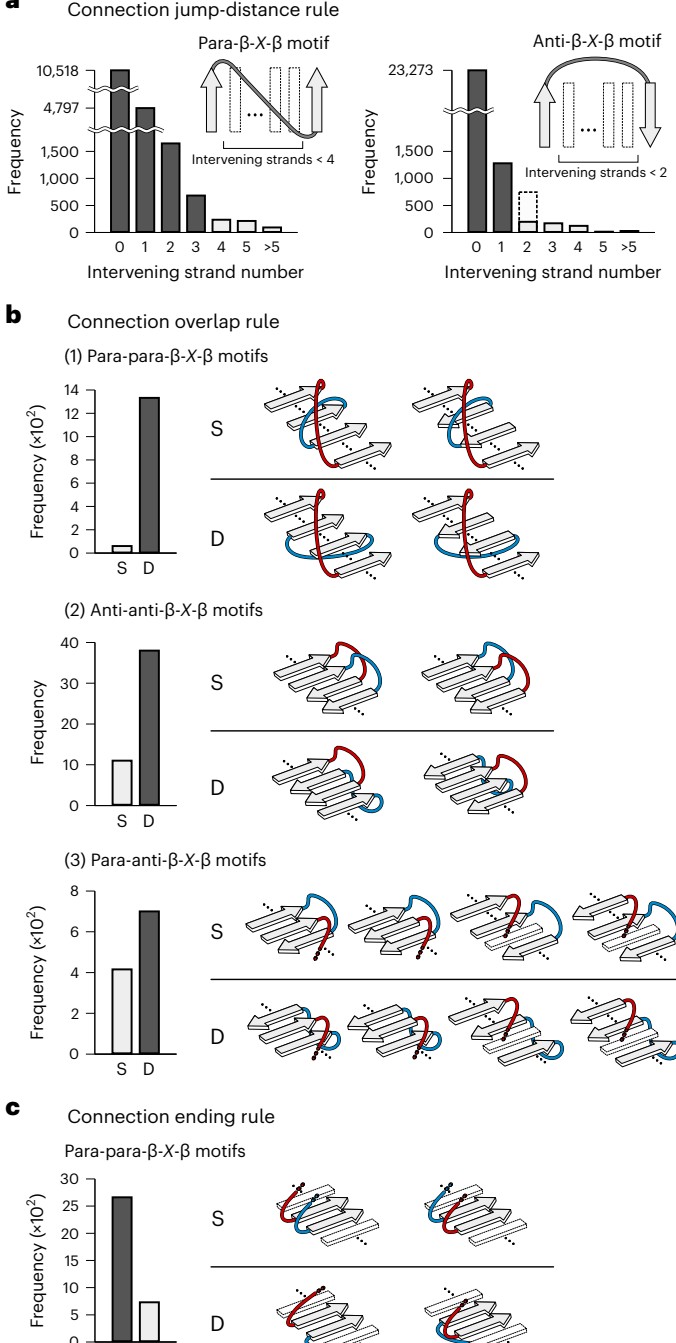

**Fig. 1 | Observed and unobserved β-sheet topologies in nature. a**, αβ-Folds defined on the basis of the β-sheet topology and Richardson's right-handed strand connections[16] shown in **b**. The upper panel shows a β-sheet topology frequently observed in nature and its corresponding ferredoxin-like fold, the lower panel shows a β-sheet topology unobserved in nature and its corresponding fold. Each β-strand is numbered according to its order along the linear chain. Gray-colored β-strand connections are on the front side of the β-sheet, black-colored ones are on the back side. **b**, Richardson's rule on the connection handedness of para-β-X-β motifs[16]. The right-handed strand connection (dark gray bar) rather than the left-handed one (light gray bar) is predominantly observed in naturally occurring proteins. **c**, Numbers of observed, unobserved and theoretically possible β-sheet topologies for each number of constituent β-strands in a β-sheet (see Fig. 3 and Methods for the definition of observed and unobserved topologies).

only those folds with right-handed connections between parallel-aligned β-strands, as per Richardson's rule[16] (Fig. 1b). This gave rise to $n! \times 2^{n-2}$ patterns in total for αβ-folds for an $n$-stranded β-sheet, including numerous αβ-folds not observed in nature (Fig. 1c). However, apparently, not all of the unobserved folds identified are possible. For example, the fold shown in the lower panel in Fig. 1a is not possible because the two β-strand connections are overlapping. Therefore, we introduced a criterion that predicts possible αβ-folds among all patterns of β-sheet topologies on the basis of a set of rules for β-sheet topology.

## Rules for β-sheet topology

We derived a set of rules from the conformational preferences of β-X-β motifs in naturally occurring protein structures, where X represents any backbone conformation (Methods): the connection jump-distance rule for single β-X-β motifs, and the connection overlap and connection ending rules for pairs of β-X-β motifs.

**Connection jump-distance rule.** 'The large number of intervening β-strands between the two β-strands is disfavored: the number of the intervening β-strands (that is, jump distance) for parallel (para) β-X-β motifs is less than four and that for antiparallel (anti) β-X-β motifs is less than two (Fig. 2a)'. An exception is the anti-β-X-β motif with two intervening β-strands included in the Greek key β-sheet topology and its circular permutations (the dotted bar in Fig. 2a and the topologies with asterisks in Fig. 3d).

**Connection overlap rule.** 'Geometrical overlap between the connections of two β-X-β motifs is less favorable: the β-sheet topologies with the two connections on the same side (S-type) are less favorable than those with the two connections on different sides (D-type) (Fig. 2b)'. Para-β-X-β motifs have a right-handed connection preference

**Fig. 2 | Rules for β-sheet topology. a**, Connection jump-distance rule. The jump distance is the number of intervening β-strands between the two β-strands of β-X-β motifs. Para-β-X-β motifs with jump distances of three or less and anti-β-X-β motifs with jump distances of one or less are frequently observed compared to β-X-β motifs with larger jump distances. The same preferences have been previously reported[49]. We revisited them using the current PDB data. **b**, Connection overlap rule. D-type β-sheet topologies (loops are located on different sides) are more frequently observed than S-type topologies (loops are located on the same side). Blue- and red-colored motifs indicate two different β-X-β motifs. Similar rules have been reported for para-para-β-X-β motifs[21,50]. For anti-anti-β-X-β motifs, a rule termed 'pretzels' has been reported[20,21], but this rule prohibits both S- and D-types. **c**, Connection ending rule. S- and D-types of β-sheet topologies for pairs of para-β-X-β motifs, in which the second strands of the two motifs are adjacent and parallel-aligned, are shown. S-type β-sheet topologies are more frequently observed than D-type topologies.

according to Richardson's rule (Fig. 1b). Analysis of anti-β-*X*-β motifs in naturally occurring protein structures revealed that the connections in anti-β-*X*-β motifs with a jump-distance number of one preferentially have a right-handed bending orientation (Extended Data Fig. 1). These right-handed connection preferences led to the connection overlap rule (Fig. 2b).

**Connection ending rule.** 'When the second strands in two para-β-*X*-β motifs are adjacent to each other and aligned in parallel, the β-sheet topologies with the two connections ending on the same β-sheet side (S-type) are preferred over those with the connections ending on different β-sheet sides (D-type) (Fig. 2c)'. Analysis of para-β-*X*-β motifs revealed that register shifts between the second strand in a para-β-*X*-β motif and the adjacent parallel-aligned β-strands are nearly always zero or positive[17] (Extended Data Fig. 2). In addition, we previously described the αβ rule: the vector from the Cα to Cβ atoms of the first strand residue following the loop connecting the helix to the strand points away from the helix[8]. These two preferences led to the connection ending rule (Fig. 2c and Extended Data Fig. 3).

### Prediction of nonfrustrated novel four-stranded αβ-folds

Using the set of rules for β-sheet topology, we classified all open β-sheet topologies with three to eight strands into frustration-free ones, without violations of the rules, and frustrated ones, with violations. We regarded frustration-free topologies as possible topologies. Many of the observed αβ-folds were identified as frustration-free, while most of the unobserved and scarcely observed αβ-folds, were identified as frustrated (Fig. 3a,b; Methods). Moreover, the frustration-free β-sheet topologies were observed in more homologous groups (that is, evolutionarily independent groups, which are referred to as superfamilies in SCOP[2] and CATH[3]) than the frustrated ones (Fig. 3c; Methods). These results suggest that the set of rules allows distinguishing possible β-sheet topologies among all β-sheet topologies.

The 96 patterns of the frustrated and frustration-free β-sheet topologies of four-stranded αβ-proteins are shown in Fig. 3d. About half of the topologies (53 patterns) were frustrated, 37 of which are either unobserved or very rare in nature. For example, the frustrated topology in column 1342 (strand order), row ↑↓↓↓ (strand orientation), which violates the connection jump distance and connection overlap rules (violations are indicated in red), has not been observed in nature. The other half (43 patterns) were frustration-free β-sheet topologies, 35 of which have been observed in nature. For example, the frustration-free β-sheet topology in column 1234, row ↑↓↑↓, termed 'meander', is the most frequently observed one. We identified eight frustration-free β-sheet topologies that have not been observed (numbered 5 to 8) or have rarely been observed (numbered 1 to 4) in nature. The latter are barely conserved in the Homology groups to which they belong. This evolutionary instability indicates the possibility that proteins with the topologies may not be robustly foldable. We regarded the αβ-folds with these eight β-sheet topologies as possible and unobserved folds (that is, novel folds) and attempted to carry out de novo design for all the predicted folds. Note that the β-sheet topology that consists of parallel-aligned β-strands with a 3142 strand order (numbered 8 in the figure) forms a knot; this topology has not been observed in nature and has long been considered to be impossible to exist[18–21]. However, we selected this topology for de novo protein design.

### De novo design of all predicted novel four-stranded αβ-folds

To evaluate whether or not the predicted novel αβ-folds can be created, we carried out de novo design of αβ-fold proteins with the eight predicted novel four-stranded β-sheet topologies (Fig. 4a,b). The αβ-folds were named NF1 to NF8 according to the order of the observation frequencies of their β-sheet topologies; NF1 to NF4 have been scarcely observed, and NF5 to NF8 have never been observed in nature (NF6–NF8 have been reported as unobserved folds[20]). We sought to design the novel αβ-folds with ideal and simple structures, in which the secondary structures do not have β-bulges or α-helix kinks and the *X* region in para-β-*X*-β motifs is an α-helix. For each αβ-fold, we built a backbone blueprint, in which secondary structure lengths and loop ABEGO torsion patterns ('A' indicates the alpha region of the Ramachandran plot, 'B', the beta region, 'G' and 'E', the positive phi region and 'O', the *cis*-peptide conformation[9,22]) were specified using backbone design rules[8,9] so that the target fold was favored (Fig. 4b). For NF1, 3, 4, 5 and 7, α-helices were appended at the termini to make the hydrophobic cores sufficiently large. For the same reason, the *X* region in the anti-β-*X*-β motifs of NF5, 6 and 7 were built with an α-turn motif[22], not just a single loop. In particular, for NF7, 'AAAB' loops for βα connections with the right twist angle (Extended Data Fig. 4) and 'BA' loops for αβ connections (Extended Data Fig. 5) were adopted to ensure that the two α-turns were packed together. For NF8, the knot-forming fold, two backbone blueprints were built using different torsion types for the loop immediately before the last strand (Extended Data Fig. 6).

Next, for each blueprint, we built a backbone structure by averaging over several hundreds of backbone structures[23] generated by Rosetta fragment assembly simulations[24] (Fig. 4c; see Methods for details). As described in the previous section, the β-sheet topologies of NF1 to NF4 have rarely been observed and those of NF5 to NF8 have not been observed in nature. To investigate whether similar naturally occurring protein structures exist in terms of the entire backbone level, we performed database analysis using MICAN[25,26] and TM-align[27], with visual inspection using the TOPS diagram[28]; no similar naturally occurring protein structures were found, except for the NF2 and NF4 designs (Extended Data Fig. 7). Subsequently, we built side chains on each of the generated backbone structures using the Rosetta design algorithm[13,29] (see Methods for details). Designs with low energy, tight core packing and high local sequence–structure compatibility[8] were selected, and their energy landscapes were explored by Rosetta ab initio structure prediction simulations[30]. Designs with amino acid sequences exhibiting funnel-shaped energy landscapes toward the designed structure were experimentally characterized (Fig. 4d).

---

**Fig. 3 | Distributions of frustration-free and frustrated β-sheet topologies in nature. a**, Numbers of frustration-free and frustrated β-sheet topologies in each observed or unobserved topology in nature for each number of constituent β-strands in a β-sheet. ªThe number within each bracket indicates the percentage of unobserved topologies in frustration-free topologies. **b**, Observation frequencies of all possible 96 topologies for four-stranded β-sheets sorted by frequency. The observation frequency of a topology in nature is represented by the number of homologous groups (superfamily) having the topology (see Methods for details). We regarded topologies with an observation frequency of less than 1/4, at which the slope changes substantially, as unobserved. **c**, Ratios of frustration-free and frustrated β-sheet topologies depending on the observation frequency for each number of constituent β-strands in a β-sheet. The number in each band indicates the number of each topology. The observation frequency is presented as the logarithm to base 4. **d**, Distributions of frustration-free and frustrated topologies in nature for all possible 96 topologies of four-stranded β-sheets. β-Strand order indicates in which order the β-strands, numbered along the sequence, are aligned in a β-sheet from left to right; β-strand orientation indicates orientations of the β-strands. In each grid cell, a β-sheet topology is illustrated with its observation frequency in nature indicated by the number below the topology and the background color gradient from white (low frequency) to yellow (high frequency). Frustration-free and frustrated topologies are represented in dark gray and light gray, respectively. β-Sheet topologies corresponding to the Greek key and its circular permutations are marked with an asterisk. Red-colored loops represent topologies including at least one frustration. Topologies enclosed in a bold black square and numbered from one to eight are unobserved frustration-free β-sheet topologies.

**Experimental characterization of designed proteins**

We obtained synthetic genes encoding 16 designs for NF1, four for each of NF2 and NF3, six for each of NF4–7 and twelve for NF8 (six for each of the two blueprints). All sequences are described in Supplementary Tables 1–8. For all sequences, no clear homologous proteins to any known protein were found (all designs have BLAST *E* values >10⁻³ against the NCBI nr database of nonredundant protein sequences). The proteins were expressed in *Escherichia coli* with C-terminal 6xHis-tags and purified using a Ni-NTA affinity column. In total, 56 out of the 60 designed proteins were expressed well and soluble. These were then characterized by circular dichroism (CD) spectroscopy, size-exclusion chromatography combined with multi-angle light scattering (SEC-MALS) and ¹H-¹⁵N heteronuclear single quantum coherence (HSQC) NMR spectroscopy. The experimental results for all designs for all target folds are summarized in Extended Data Table 1. The success rate of the designs including the knotted

**a**

| No. of strands | No. of total topologies | Observed topologies | | Unobserved topologies | |
|---|---|---|---|---|---|
| | | No. frustration free | No. frustrated | No. frustration free (novel)[a] | No. frustrated |
| 3 | 12 | 10 | 2 | 0 (0.0) | 0 |
| 4 | 96 | 35 | 16 | 8 (18.6) | 37 |
| 5 | 960 | 77 | 42 | 111 (59.0) | 730 |
| 6 | 11,520 | 64 | 46 | 663 (91.2) | 10,747 |
| 7 | 161,280 | 30 | 43 | 2,571 (98.8) | 158,636 |
| 8 | 2,580,480 | 18 | 17 | 9,003 (99.8) | 2,571,442 |

**b**

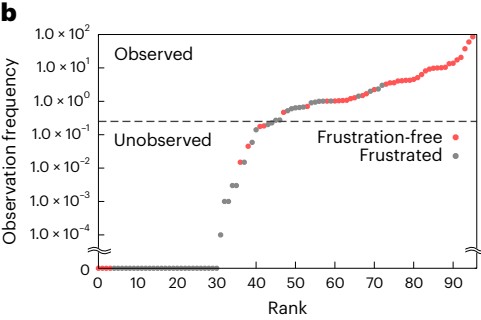

**c**

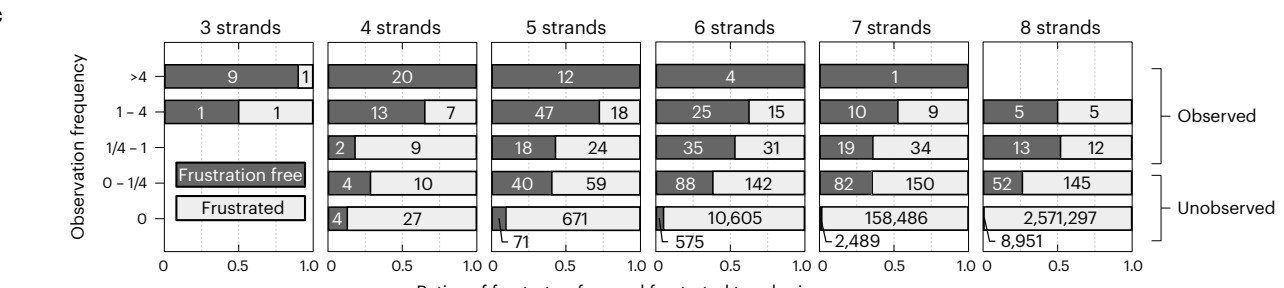

Ratios of frustration-free and frustrated topologies

**d**

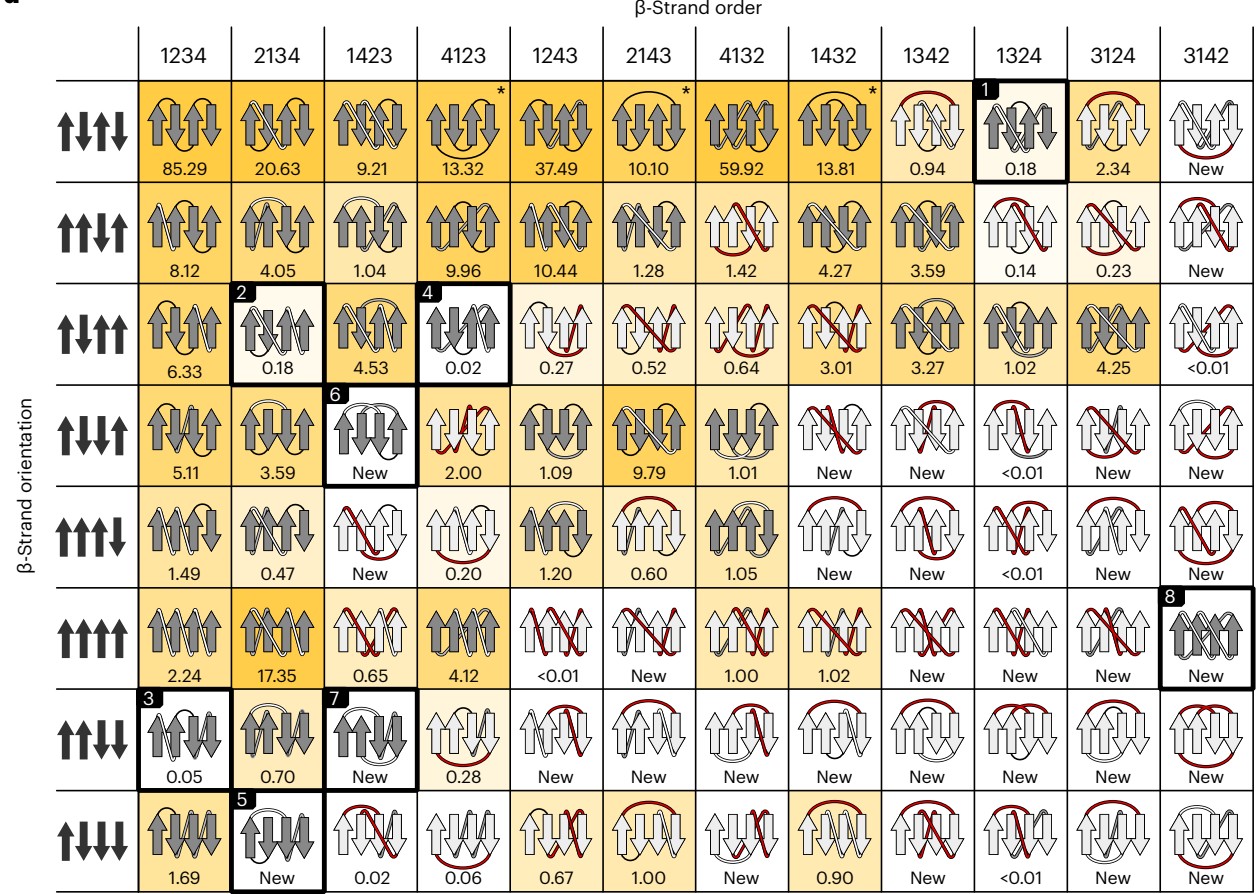

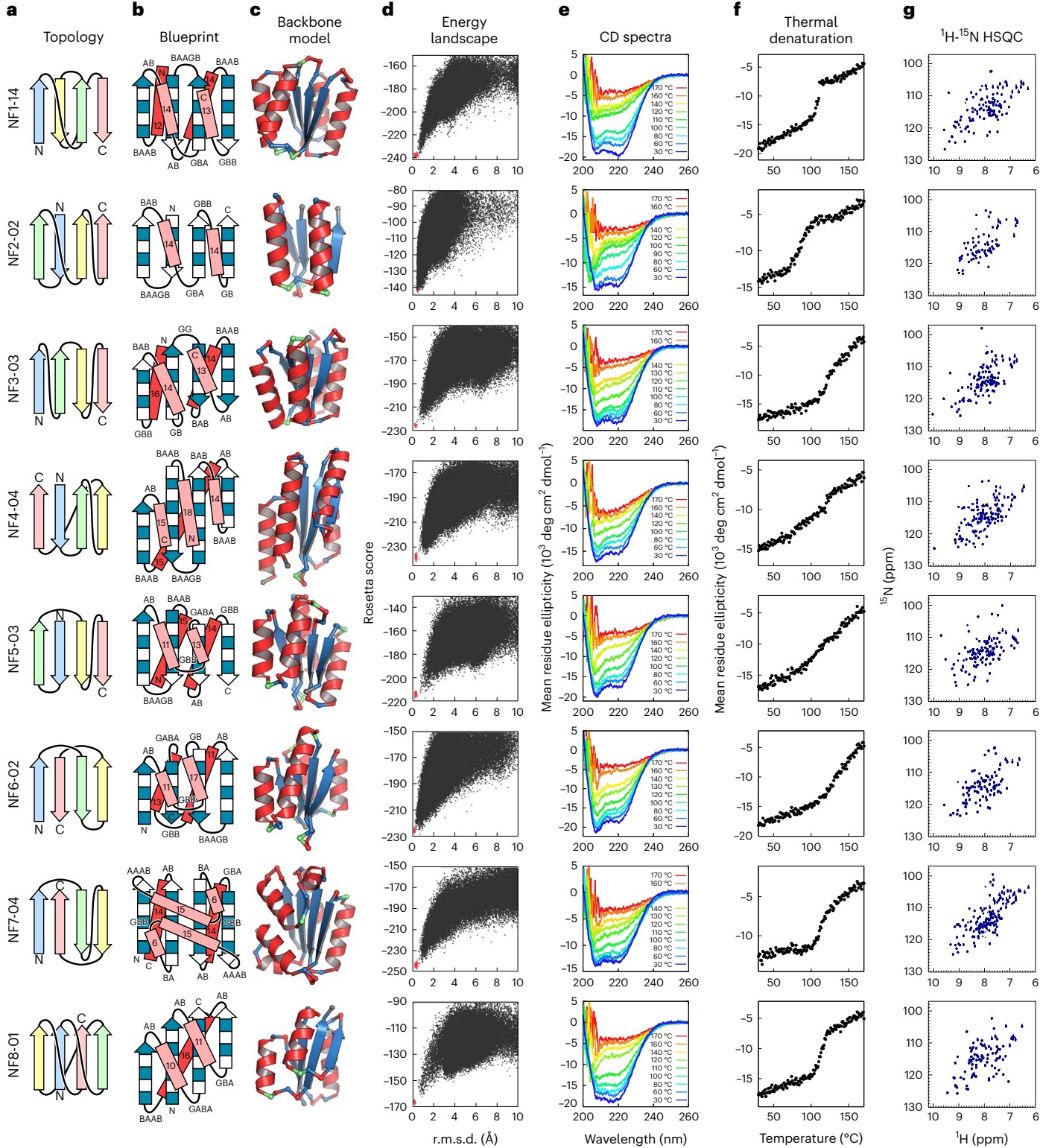

**Fig. 4 | Characterization of the designs for all eight novel αβ-folds.**
**a**, Identified novel β-sheet topologies. **b**, Backbone blueprints used for de
novo design of the novel αβ-fold structures. Strand lengths are represented
by filled and empty boxes that represent pleats coming out and going into the
page, respectively. Letter strings next to the loops indicate their ABEGO torsion
patterns[9]. **c**, Backbone structures generated from the blueprints. Each residue
color represents its ABEGO torsion angle (red, A; blue, B; green, G). **d**, Energy

landscapes obtained from Rosetta ab initio structure prediction simulations[30].
Each dot represents the lowest energy structure obtained in an independent
trajectory starting from an extended chain (black) or the design model (red) for
each sequence; the x axis shows the Cα r.m.s.d. from the design model and the y
axis shows the Rosetta all-atom energy. **e**, Far-ultraviolet CD spectra at various
temperatures (30–170 °C). **f**, Thermal denaturation monitored at 222 nm.
**g**, Two-dimensional $^{1}$H-$^{15}$N HSQC spectra at 25 °C and 600 MHz.

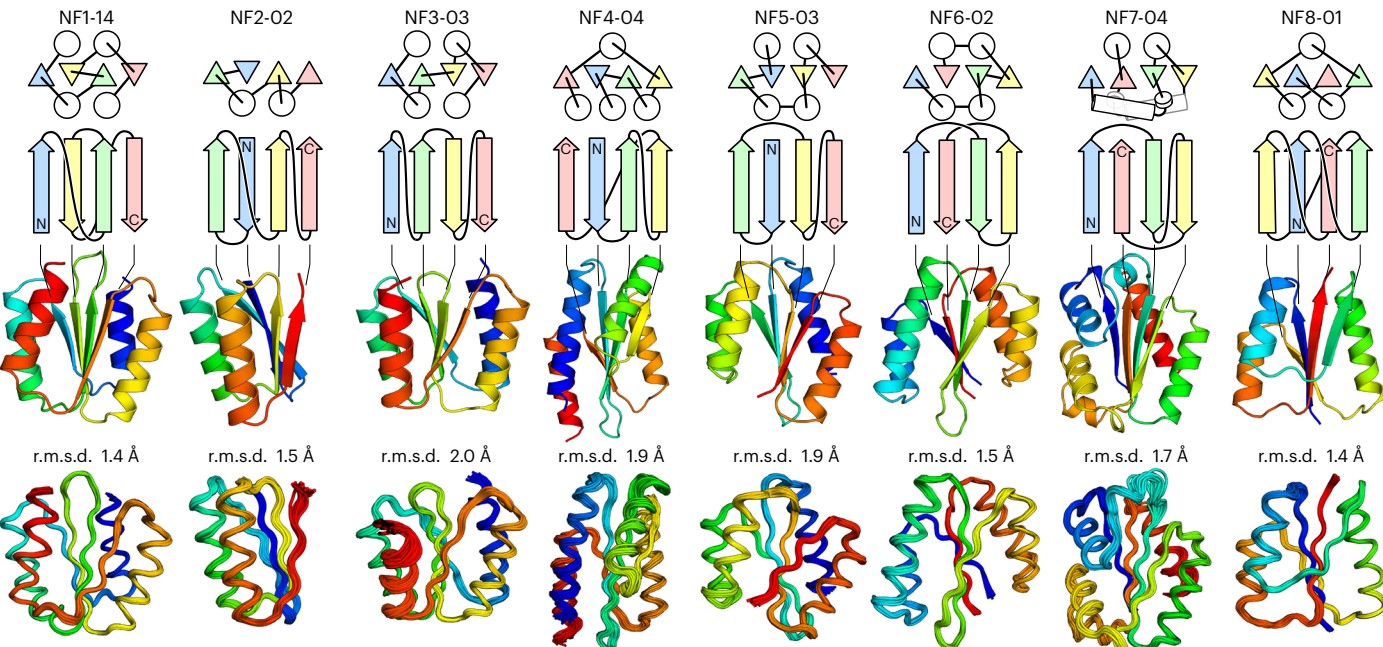

**Fig. 5 | Comparison of computational models with experimentally determined structures.** Top, the top two rows show designed novel αβ-folds from NF1 to NF8. The tertiary arrangement of α-helices (circles) and β-strands (triangles) and their connections are shown at the top, the β-sheet topologies below. Middle, computational design models. Bottom, the NMR structures. The r.m.s.d. between the design model and NMR structure for backbone heavy atoms is indicated. The design models are available in Supplementary Data 1, the NMR structures are available in the PDB: NF1-14 (PDB 7BPL), NF2-02 (7BPM), NF3-03 (7BQE), NF4-04 (7BQC), NF5-03 (7BPP), NF6-02 (7BQB), NF7-04 (7BPN) and NF8-01 (7BQD).

fold was as high as those in previous de novo designs with the folds existing widely in nature (28 out of 60 designs were characterized as foldable proteins)[8–12]. For each target fold, one monomeric design with a CD spectrum characteristic of αβ-proteins and the expected number of well-dispersed sharp NMR peaks were selected for NMR structure determination (Fig. 4e–g). All the designs exhibited high thermal stability. Interestingly, the thermal denaturation curves for the designed proteins of NF3–6 were not cooperative (Fig. 4f), which could be one of the reasons why these folds have not been observed in nature. The NMR structures solved using MagRO-NMRViewJ[31,32] (Supplementary Fig. 1) were in close agreement with the computational design models for both the backbone (Fig. 5; the root mean square deviation (r.m.s.d.) values for backbone heavy atoms ranged from 1.4 to 2.0 Å) and the core side chains (Extended Data Fig. 8 and Supplementary Table 9), with the correct β-sheet topologies (see Table 1 for NMR constraints and structure statistics). Remarkably, we succeeded in designing the smallest knotted NF8 structure consisting of only four strands (Extended Data Fig. 9). The successful de novo design of all eight αβ-folds demonstrates that the set of rules allows prediction of the novel αβ-folds.

### Prediction of novel five- to eight-stranded αβ-folds
Next, we revisited the number of frustration-free unobserved β-sheet topologies with five- to eight-stranded β-sheets, shown in Fig. 3a (for three-stranded αβ-proteins, all ten frustration-free β-sheet topologies have been observed in nature). As the number of constituent β-strands in a β-sheet increases, the number of frustration-free unobserved topologies increases exponentially and the ratio of unobserved topologies in frustration-free topologies also increases. The prediction indicates that 12,348 frustration-free (that is, possible) αβ-folds have been left as unobserved in nature; this number far exceeds that of the αβ-folds observed in nature (that is, 400 folds). Note that, since we only investigated novel folds identified by the set of rules introduced here, the predicted number corresponds to a lower limit of that of novel folds.

There must be more novel folds that are not identified by applying the rules, but accessible to polypeptide chains.

## Discussion
The extent of the protein fold space that is accessible to the polypeptide chain has long been unclear. We systematically investigated the unexplored fold space by introducing a set of rules to predict novel αβ-folds and by carrying out de novo design of all the predicted novel αβ-folds with a four-stranded β-sheet. We found that all the predicted novel αβ-folds, including a knotted fold, can be created. Remarkably, the design success rate was comparable to that of previous de novo designs with naturally occurring folds, and the thermal stability of the designs was as high as that of previous designs[8–12]. Our study indicates that there are more than 10,000 novel αβ-folds with five- to eight-stranded β-sheets.

There are several possible reasons for why these novel folds have not been observed in nature: (1) all life on Earth descended from a common ancestor: naturally occurring folds have been repetitively reused and adapted for different functions, and, therefore, life on Earth is biased by this ancestral relationship and the novel folds have, by chance, not emerged; living things could have evolved using a set of protein folds different from that currently observed in nature; (2) the timeline of biological evolution so far is too short for all possible folds to be explored; and (3) the novel folds are incapable of carrying out functions required for life and have therefore become extinct due to evolutionary bias. To address these possibilities, the relationship between novel fold structures and their functions need to be studied.

We tested whether AlphaFold2 (ref. 33) can predict the designed structures with novel αβ-folds from their amino acid sequences, using the template structure database as it existed before 14 May 2009 when many de novo-designed proteins started to be deposited in the PDB; the designed structures for NF2, 4, 5 and 6 were predicted for all five prediction models, but those for NF1, 3, 7 and 8 were not predicted

**Table 1 | NMR constraints and structure statistics of the eight designed structures**

| Design identity | NF1-14 (PDB 7BPL), (BMRB 36327) | NF2-02 (PDB 7BPM), (BMRB 36328) | NF3-03 (PDB 7BQE), (BMRB 36334) | NF4-04 (PDB 7BQC), (BMRB 36332) |
|---|---|---|---|---|
| **NMR distance and dihedral constraints** | | | | |
| Distance constraints | | | | |
| Total NOE | 2,598 (100.0%) | 1,383 (100.0%) | 2,160 (100.0%) | 1,944 (100.0%) |
| Intra-residue | 525 (20.2%) | 319 (23.1%) | 529 (24.5%) | 493 (25.4%) |
| Inter-residue | | | | |
| Sequential ($\lvert i-j \rvert=1$) | 672 (25.9%) | 391 (28.3%) | 565 (26.2%) | 481 (24.7%) |
| Medium range ($1<\lvert i-j \rvert<5$) | 585 (22.5%) | 341 (24.7%) | 498 (23.1%) | 375 (19.3%) |
| Long range ($\lvert i-j \rvert\geq5$) | 816 (31.4%) | 332 (24.0%) | 568 (26.3%) | 595 (30.6%) |
| Total dihedral angle restraints | 167 | 114 | 178 | 200 |
| $\phi$ | 83 | 57 | 89 | 100 |
| $\psi$ | 84 | 57 | 89 | 100 |
| **Structure statistics** | | | | |
| Violations (mean and s.d.)[a] | | | | |
| Distance constraints (Å) | 0.2±0.01 (17) | 0.03±0.06 (4) | 0.017±0.04 (15) | 0.019±0.07 (2) |
| Dihedral angle constraints (°) | 12.3±14.3 (10) | 11.8±7.6 (15) | 5.9±10.9 (2) | 13.12±9.2 (2) |
| Max. distance constraint violation (Å) | 0.22 | 0.14 | 0.28 | 0.19 |
| Max. dihedral angle violation (°) | 28.7 | 18.1 | 59.7 | 43.51 |
| Deviations from idealized geometry[b] | | | | |
| Bond lengths (Å) | 0 | 0 | 0 | 0 |
| Bond angles (°) | 0 | 0 | 0 | 0 |
| Impropers (°) | 0 | 0 | 0 | 0 |
| Average pairwise r.m.s.d.[c] (Å) | | | | |
| Heavy | 0.96±0.10 | 1.29±0.11 | 1.29±0.12 | 1.39±0.14 |
| Backbone | 0.30±0.05 | 0.40±0.07 | 0.58±0.09 | 0.65±0.13 |
| RDC validation[d] | | | | |
| Total number of RDC values | 80 | 52 | 99 | 78 |
| $R_P^{free}$ | 0.904±0.006 | 0.923±0.006 | 0.920±0.008 | 0.917±0.012 |
| $Q^{free}$ (%) | 33.8±1.0 | 37.7±1.0 | 32.6±2.4 | 38.0±2.6 |

| Design identity | NF5-03 (PDB 7BPP), (BMRB 36330) | NF6-02 (PDB 7BQB), (BMRB 36331) | NF7-04 (PDB 7BPN), (BMRB 36329) | NF8-01 (PDB 7BQD), (BMRB 36333) |
|---|---|---|---|---|
| **NMR distance and dihedral constraints** | | | | |
| Distance constraints | | | | |
| Total NOE | 1,943 (100%) | 1,802 (100.0%) | 1,902 (100.0%) | 1,654 (100.0%) |
| Intra-residue | 502 (25.8%) | 467 (25.9%) | 525 (27.6%) | 402 (24.5%) |
| Inter-residue | | | | |
| Sequential ($\lvert i-j \rvert=1$) | 458 (23.6%) | 457 (25.4%) | 473 (24.9%) | 449 (27.1%) |
| Medium range ($1<\lvert i-j \rvert<5$) | 380 (19.6%) | 397 (22.0%) | 351 (18.5%) | 326 (19.7%) |
| Long range ($\lvert i-j \rvert\geq5$) | 603 (31.0%) | 481 (26.7%) | 553 (29.1%) | 477 (28.8%) |
| Total dihedral angle restraints | 187 | 183 | 214 | 127 |
| $\phi$ | 93 | 89 | 107 | 63 |
| $\psi$ | 94 | 90 | 107 | 64 |
| **Structure statistics** | | | | |
| Violations (mean and s.d.)[a] | | | | |
| Distance constraints (Å) | 0.001 (1) | 0.0 (0) | 0.001 (1) | 0.01 (1) |
| Dihedral angle constraints (°) | 2.9±2.8 (12) | 9.28±2.0 (12) | 25.2±7.3 (22)[e] | 2.42 (1) |
| Max. distance constraint violation (Å) | 0.18 | N/A | 0.19 | 0.20 |
| Max. dihedral angle violation (°) | 50.0 | 38.45 | 51.9 | 48.4 |

**Table 1 (continued) | NMR constraints and structure statistics of the eight designed structures**

| Design identity | NF5-03 (PDB 7BPP), (BMRB 36330) | NF6-02 (PDB 7BQB), (BMRB 36331) | NF7-04 (PDB 7BPN), (BMRB 36329) | NF8-01 (PDB 7BQD), (BMRB 36333) |
|---|---|---|---|---|
| Deviations from idealized geometry[b] | | | | |
| Bond lengths (Å) | 0 | 0 | 0 | 0 |
| Bond angles (°) | 0 | 0 | 0 | 0 |
| Impropers (°) | 0 | 0 | 0 | 0 |
| Average pairwise r.m.s.d.[c] (Å) | | | | |
| Heavy | 1.37±0.10 | 1.42±0.11 | 1.57±0.16 | 1.24±0.09 |
| Backbone | 0.52±0.07 | 0.62±0.10 | 0.69±0.13 | 0.48±0.07 |
| RDC validation[d] | | | | |
| Total number of RDC values | 89 | 78 | 98 | 57 |
| $R_P^{free}$ | 0.908±0.010 | 0.955±0.007 | 0.915±0.010 | 0.931±0.012 |
| $Q^{free}$ (%) | 41.2±2.0 | 31.9±2.7 | 40.4±2.2 | 31.6±2.8 |

[a]Mean and s.d. values are derived from 20 models of Amber refined structures. Total number of violations found in ordered residues of all models for dihedral angle constraints (<20°) and for distance constraints (<0.2 Å) are indicated in parentheses. For distance constraints of NF5-03, NF7-04 and NF8-01, and dihedral angle constraints of NF8-01, only one violation is found for only one model, so s.d. is not estimated. [b]No geometrical outliers are found in all models. [c]Averaged r.m.s.d. and deviation of backbone and heavy atoms for all pair of models in ensemble (20×20) are calculated by MolMol[46], fitted on the residues in ordered region (NF1-14: 1–83, 86–103; NF2-02: 1–67; NF3-03: 2–101; NF4-04: 2–71, 79–81, 91–97, 101–111; NF5-03: 2–100; NF6-02: 2–97; NF7-04: 3–8, 28–37, 56–61, 94–114; NF8-01: 1–80) identified by Filt_Robot[47]. [d]Experimentally obtained $^1H$-$^{15}N$ residual dipolar coupling (RDC) values ($^1D_{1H/15N}$) are exclusively used for the RDC validation of NMR models (Methods). The Pearson's correlation coefficient $R_P^{free}$ and the Cornilescu $Q^{free}$ are calculated by PALES[48]. [e]Violations are mainly found in the dihedral angle constraints of Asn11 or Leu29.

for all the models. The prediction method relies on the information obtained from the evolutionary history of naturally occurring proteins; predictions for amino acid sequences far from the ones in nature could be difficult.

The number of predicted novel αβ-folds, which is at the lower limit of that of novel αβ-folds, far exceeds that of the folds observed in nature. Moreover, the novel αβ-folds include the knot-forming ones. Recently, functional proteins have been designed de novo[34–45]. The novel αβ-folds predicted in this study should provide a vast scaffold set for designing protein structures with desired functions.

## Online content

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

## Methods

### Structure dataset of naturally occurring proteins

For the derivation of a set of rules for β-sheet topology (Fig. 2), a dataset comprising 12,595 chains obtained from the cullpdb database (accessed 13 December 2018)[51] with more than 40 residues, sequence identity <25%, resolution <2.5 Å and *R*-factor <1.0 was used. For the analysis of β-sheet topologies of naturally occurring protein structures (Fig. 3), a dataset comprising 65,371 domains obtained from the semimanually curated domain database ECOD[52], which provides a hierarchical grouping of evolutionarily related domains, with more than 40 residues and sequence identity <99%, was used. For all obtained structures, structure refinements were performed using ModRefiner[53], and the secondary structures were assigned using STRIDE[54]; when the r.m.s.d. of a refined structure against the original structure for Cα atoms was >1.0 Å, the refined structure was discarded and the original one was used.

### Analysis of β-sheet topologies in naturally occurring proteins

β-Sheet topologies were defined for open β-sheets included in the protein domains obtained from ECOD on the basis of the following criteria: (1) the lengths of constituent β-strands are more than two residues; (2) the number of β-strands is at least three; (3) two neighboring β-strands have at least two main chain hydrogen bonds between the β-strands; and (4) no insertion along a sequence by any β-strands belonging to another β-sheet consisting of more than two β-strands (Supplementary Fig. 2). Branched β-sheets with β-strands having more than two neighboring β-strands were discarded.

The observation frequencies in nature of all β-sheet topologies were studied using the ECOD database, in which protein domains are classified according to their evolutionary relationships. In the database, two categories, Family and Homology, are defined. Family represents a group of evolutionarily related protein domains identified on the basis of substantial sequence similarity, and Homology represents a group comprising multiple Family groups, evolutionary relationships of which are inferred on the basis of functional and structural similarities (Homology is equivalent to the superfamily in the SCOP[2] or CATH[3] structure databases). To study the observation frequency for each β-sheet topology, we counted the Homology groups having the topology, with the following consideration. We first examined the occupation ratio (OR) of the topology in the *i*th Homology group:

$$OR(i) = \frac{\sum_{j}^{N_{Family}} R_{Family}(j)}{N_{Family}},$$

where $N_{Family}$ is the total number of Family groups belonging to the Homology group and $R_{Family}(j)$ is the ratio of protein domains having the β-sheet topology in the *j*th Family group. Thus, when all domains in the Homology group contain the β-sheet topology, the occupation ratio of the Homology group is one; otherwise, it is less than one. Finally, the observation frequency for each topology is calculated as the sum of the occupation ratios across Homology groups:

$$\sum_{i}^{N_{Homology}} OR(i).$$

For four-stranded β-sheet proteins, we manually checked all structures having topologies with observation frequencies <1.0 and then changed the β-sheet assignments for some of the structures: the β-sheets included in e3hy2X1, e1xw3A1, e2hwjA2, e4rsfA1 and e1tocR2 were identified as β-barrels, and those in e4rgzA1, e2bjjX3, e1iejA2, e3s9lC3, e1blfA4 and e2d3iA3 were identified as six-stranded β-sheets. The defined observation frequency was used to distinguish observed and unobserved topologies in this study: topologies with the observation frequency of 0 were considered unobserved, and evolutionarily unstable topologies with the observation frequency of less than 1/4 were also considered unobserved (Fig. 3b).

### Backbone construction

We built a backbone blueprint for each novel αβ-fold. For the *X* region in para-β-*X*-β motifs, a helix was built. The lengths of secondary structures and ABEGO torsion patterns for the connecting loops were obtained from previously reported design rules[9]. For NF1, 3, 4, 5 and 7, α-helices were appended to their termini to ensure a sufficiently large hydrophobic core between the β-sheet and the α-helices. For the same purpose, an α-turn structure consisting of a helix–loop–helix unit was built in the *X* region in antiparallel β-*X*-β motifs in NF5, 6 and 7. β-Strand lengths were selected from 4 to 7, and α-helix lengths varied from 11 to 17 residues. The torsion ABEGO patterns of loops were as follows: GB, GBA or BAAB for the connecting loops of para-type αβ units, AB for para-type βα units, BAB or GBB for anti-type βα units, BAAGB for *R* chiral ββ units and GG for *L* chiral ββ units. We newly introduced GABA for para-type αβ units and GBB for anti-type αβ units (Extended Data Fig. 5). For α-turn structures, we used the GBB loop[22] to connect the two helices. In the NF7 blueprint, AAAB for anti-type βα units (Extended Data Fig. 4) and BA for para-type αβ units (Extended Data Fig. 5) were used to arrange the two α-turns packed with each other.

In total, 1,000–40,000 backbone structures for each blueprint (sufficient number depends on its fold type) were generated by Rosetta sequence-independent Monte Carlo fragment assembly simulations using coarse-grained model backbone structures, in which each residue is represented by main chain atoms (N, H, CA, C and O) and a side chain pseudo atom[30]. The Rosetta potential function used in the simulations considers steric repulsion (vdw = 1.0), overall compaction (rg = 1.0), secondary structure pairings (ss_pair = 1.0, rsigma = 1.0 and hs_pair = 1.0) and main chain hydrogen bonds (hbond_sr_bb = 1.0 and hbond_lr_bb = 1.0), with no sequence-dependent score terms. The steric radius of Val was used for that of the side chain pseudo atom. The ss_pair and rsigma score terms were modified so that only the strand residue pairs specified in the blueprint were favored in the simulations. To enhance the sampling efficiency for obtaining target topology backbone structures, we built backbone structures part by part. For instance, for the NF2 fold, the N-terminal half (β1–β2–α1–β3), which forms a locally globular substructure, was built first, and, subsequently, the C-terminal half (α2–β4) was built by extending the N-terminal half. The generated backbone structures were further refined as follows. (1) β-Sheet refinement. The entire structure was minimized with constraints making the Cα atoms of the neighboring strand residues in the blueprint to be <5.5 Å, using the Rosetta full-atom FastRelax protocol[55] with upweighted hydrogen bonding and backbone torsion angle terms (hbond_sr_bb = 5.0, hbond_lr_bb = 3.0 and omega = 3.0). Val was used for the full-atom side chains for all residues, except for those in the G region in the ABEGO Ramachandran map[22] (for which Gly was used). This step was repeated up to ten times until the secondary structures and ABEGO torsion patterns became identical to those designated in the blueprint. (2) α-Helix refinement. The loop–helix–loop structures were rebuilt using the cyclic coordinate descent loop closure method[56] implemented in the BlueprintBDR mover. This step was repeated up to ten times for each loop–helix–loop region until the α-helix was built without kinks and the loop torsion patterns were identical to those designated in the blueprint. Next, we selected 100–500 backbone structures in which the terminal α-helices are packed with the β-sheet with the criterion that at least one residue in any continuous five-residue segments in the terminal α-helices is buried (accessible surface area <40 Å²) by contacting with the central two β-strands in the β-sheet. Some of the generated backbone structures showed structural diversity. In such cases, we clustered the backbone structures on the basis of structural similarity, using a hierarchical clustering approach (average linkage). The structural similarity was evaluated with Cα r.m.s.d., using a cutoff for clustering of 1.0–2.0 Å, according to the structural diversity of the generated structures. From the top three largest clusters, we selected the cluster consisting of structures with tightly packed secondary structures. Finally, we averaged the *xyz* coordinates

of the main chain atoms of 30–150 backbone structures in the cluster, followed by the Rosetta idealization protocol with upweighted score terms (hbond_sr_bb = 10.0, hbond_lr_bb = 10.0 and omega = 10.0), resulting in a backbone structure to be used for the subsequent side chain design.

## Sequence design

We performed RosettaDesign calculations[29] using the full-atom Talaris2014 (ref. [57]) scoring function to design side chains (amino acid sequences) that stabilize each generated backbone structure. The design calculation consists of the following three steps: (1) several cycles of amino acid sequence optimization with a fixed backbone and subsequent backbone relaxation; (2) mutations of buried polar residues to hydrophobic ones, followed by optimization of the entire structure; and (3) mutations of solvent-exposed hydrophobic residues to polar residues, followed by optimization of the entire structure. Amino acid types to be used for the design of each residue position, except for that of loop regions, were restricted on the the basis of the secondary structure of the position and the buriedness calculated using virtual amino acids. For the design of each loop region (the residues in the loop and the preceding and following three residues), amino acid types were restricted on the basis of the consensus amino acids obtained from the sequence profile for naturally occurring protein structure fragments, which were collected based on the following criteria: (1) secondary structure and ABEGO torsion pattern identical to those of the loop region and (2) r.m.s.d. against the loop structure <2.0 Å. Through the RosettaDesign calculations, up to 40,000 designs were generated for each design target structure.

The designed sequences were then filtered on the basis of the Rosetta total energy, RosettaHoles score[58] <2.0 and packstat score of >0.55 for NF2 and >0.6 for the others. Furthermore, we filtered the designs on the basis of the local sequence–structure compatibility[8]. We collected 200 fragments for each nine-residue frame in each designed sequence from a nonredundant set of experimental structures, on the basis of the sequence similarity and secondary structure prediction. Subsequently, for each frame, we calculated Cα r.m.s.d. of the local structure against each of the 200 fragments. The designs were ranked according to the summation of the log ratio of the fragments, for which the r.m.s.d. was <1.5 Å across all nine-residue frames, and those with high values were selected.

## Protein expression and purification

A spacer was added at the C terminus of each designed sequence ('GSWS' for the sequences that have neither a Trp residue nor more than two Tyr residues and 'GS' for others) to separate the designed region from the C-terminal 6xHis-tag. Genes encoding the designed sequences were synthesized and cloned into pET21b expression vectors at Eurofins Genomics. The designed proteins were expressed in *E. coli* BL21 Star (DE3) cells (Invitrogen) as uniformly (U)[15]N-labeled proteins using MJ9 minimal medium[59] containing [[15]N]ammonium sulfate as the sole nitrogen source and [[12]C]glucose as the sole carbon source. The expressed proteins with a C-terminal 6xHis-tag were purified using an Ni-NTA affinity column. The purified proteins were dialyzed against PBS buffer (137 mM NaCl, 2.7 mM KCl, 10 mM Na$_2$HPO$_4$ and 1.8 mM KH$_2$PO$_4$, pH 7.4; this buffer was used for all experiments except NMR structure determination). The expression, solubility and purity of the designed proteins were assessed by SDS–PAGE and mass spectrometry (Thermo Scientific Orbitrap Elite). The protein concentrations were determined from the absorbance at 280 nm (ref. [60]) measured using a UV spectrophotometer (NanoDrop, Thermo Scientific).

## Circular dichroism spectroscopy

CD data were collected on a JASCO J-1500 CD spectrometer using a JASCO SpectraManager software v.2. For all designs, far-UV CD spectra were measured from 260 to 200 nm using ~20-μM protein samples in PBS buffer (pH 7.4) with a 1-mm path length cuvette. For the eight representative designs (NF1-14, NF2-02, NF3-03, NF4-04, NF5-03, NF6-02, NF7-04 and NF8-01), thermal denaturation measurements were performed once from 30 to 170 °C under 1 MPa pressure with an increase of 1 °C min$^{-1}$. During the denaturation, the ellipticity at 222 nm was monitored, and far-UV CD spectra were measured from 260 to 200 nm at the various temperatures shown in Fig. 4e.

## Size-exclusion chromatography combined with multi-angle light scattering

SEC-MALS experiments were performed using a miniDAWN TREOS static light scattering detector (Wyatt Technology) combined with a high-performance liquid chromatography (HPLC) system (1260 Infinity LC, Agilent Technologies). One hundred microliters of 200–500 μM Ni-purified protein samples in PBS buffer (pH 7.4) was injected into a Superdex 75 Increase 10/300 GL (GE Healthcare) or Shodex KW-802.5 (Showa Denko K.K.) column equilibrated with PBS buffer at a flow rate of 0.5 ml min$^{-1}$. The protein concentrations were calculated from the absorbance at 280 nm detected by the HPLC system. Static light scattering data were collected at three different angles of 43.6°, 90.0° and 136.4° at 659 nm. The data were analyzed using ASTRA software (v.6.1.2, Wyatt Technology) with a change in the refractive index with concentration, a d$n$/d$c$ value, 0.185 ml g$^{-1}$.

## Two-dimensional $^1$H-$^{15}$N heteronuclear single quantum coherence measurement by nuclear magnetic resonance

Two-dimensional (2D) $^1$H-$^{15}$N HSQC NMR experiments were performed to verify whether the designed proteins fold into well-packed structures. The HSQC spectra were collected for protein samples of 0.5–1.0 mM in 90% $^1$H$_2$O/10% $^2$H$_2$O PBS buffer (pH 7.4) at 25 °C on a JEOL JNM-ECA 600 MHz spectrometer using Delta v.5.0.4 NMR software. The stable monomeric design with the expected number of well-dispersed sharp NMR spectra for each fold (NF1-14, NF2-02, NF3-03, NF4-04, NF5-03, NF6-02, NF7-04 and NF8-01) was selected for NMR structure determination.

## Solution structure determination by NMR

**Sample preparation.** For NMR structure determination of the eight selected designs, uniformly isotope-labeled [U-$^{15}$N, U-$^{13}$C] proteins were expressed using the method described above, except that [$^{13}$C] glucose was used as a sole carbon source. The [U-$^{15}$N, U-$^{13}$C]-enriched proteins were purified through a Ni-NTA affinity column followed by gel filtration chromatography on an ÄKTA Pure 25 FPLC (GE Healthcare) using a Superdex 75 Increase 10/300 GL column (GE Healthcare). The purified proteins were dissolved in 95% $^1$H$_2$O/5% $^2$H$_2$O PBS buffer at various pH (50 mM NaCl, 1.1 mM Na$_2$HPO$_4$ and 7.4 mM KH$_2$PO$_4$ at pH 6.0 for NF2-02, NF3-03 and NF6-02; 50 mM NaCl, 4.3 mM Na$_2$HPO$_4$ and 5.7 mM KH$_2$PO$_4$ at pH 6.8 for NF5-03, NF7-04 and NF8-01; 50 mM NaCl, 5.6 mM Na$_2$HPO$_4$ and 1.1 mM KH$_2$PO$_4$ at pH 7.4 for NF1-14; and 137 mM NaCl, 1.1 mM Na$_2$HPO$_4$ and 7.4 mM KH$_2$PO$_4$ at pH 7.4 for NF4-04). Shigemi micro-NMR tubes were used for all NMR measurements except RDC (protein concentration ~900 μM for all designed proteins except NF4-04 (~400 μM) and NF6-02 (~700 μM)), and normal NMR tubes were used for RDC experiments (protein concentration ~200 μM).

**NMR measurements.** NMR measurements were performed on Bruker AVANCE III NMR spectrometers equipped with QCI cryo-Probe ($^1$H/$^{13}$C/$^{15}$N/$^{31}$P) at 303 K. Spectrometers with 600-, 700- and 800-MHz magnets were used for signal assignments and NOE-related measurements, whereas those with 900- and 950-MHz magnets were used for RDC experiments. For signal assignments, 2D $^1$H-$^{15}$N HSQC (echo/ anti-echo), $^1$H-$^{13}$C constant-time HSQC for aliphatic and aromatic signals and three-dimensional (3D) HNCO, HN(CO)CACB and 3D HNCACB for backbone signal assignments, were measured, whereas the BEST pulse sequence[61] was used for triple-resonance experiments of NF2,

NF3, NF5, NF6, NF7 and NF8. For structure determination, 3D $^{15}$N-edited NOESY, and 3D $^{13}$C-edited NOESY for aliphatic and aromatic signals (mixing time = 100 ms), were performed. For RDC experiments, 2D IPAP $^1$H-$^{15}$N HSQC NMR using WATERGATE pulses for water suppression were measured with or without 6–10 mg ml$^{-1}$ of Pf1 phage (ASLA Biotech). To confirm the positions of $^1$H-$^{15}$N signals in the 2D IPAP $^1$H-$^{15}$N HSQC, 3D HNCO in the identical buffer condition containing Pf1 phage were measured. The α and β states of $^{15}$N signals split by $^1$H-$^{15}$N $^1J$ coupling were separately identified for the protein in the isotropic and weakly aligned states, to obtain one-bond RDC $^1D_{1H/15N}$ values. They were estimated by simple subtraction of the shifted values between isotropic and weakly aligned states then divided by the static magnetic field to obtain the RDC value in Hz.

**NMR signal assignments.** All NMR signals were identified using MagRO-NMRViewJ (upgraded version of Kujira)[31] in a fully automated manner, then noise peaks were filtered by deep-learning methods using Filt_Robot[32]. The FLYA module was used for fully automated signal assignments and structure calculation[62] to obtain roughly assigned chemical shifts (ACS) and trustful ones were selected into a MagRO ACS table. After confirmation and correction of the ACS by visual inspection on MagRO, TALOS+[63] calculations were performed to predict phi/psi dihedral angles, which were then converted to angle constraints for the CYANA format. The signal assignments in 2D $^1$H-$^{15}$N HSQC spectra for all folds are shown in Supplementary Figs. 3–10.

Before measuring a series of 3D spectra for the side chain chemical shift assignments for aliphatic and aromatic $^1$H/$^{13}$C signals, we inspected 2D $^1$H/$^{15}$N and $^1$H/$^{13}$C HSQC spectra to evaluate how many crowded, overlapped or missing signals were in these 2D spectra, and then decided the following set of 3D spectra: 3D HCCH-TOCSY, $^{13}$C-edited NOESY. The side chain amide signals were assigned using 3D H(CCO)NH, (H)C(CO)NH and $^{15}$N-edited NOESY. The details for 3D spectra are described as follows. 3D HCCH-TOCSY: for aliphatic, offset place on $^{13}$C-aliphatic center, DIPSI-3 mixing for $^{13}$C spin-lock, States-TPPI for $^1$H, $^{13}$C indirect; for aromatic, offset place on $^{13}$C-aromatic center, DIPSI-3 mixing for $^{13}$C spin-lock, States-TPPI for $^1$H, $^{13}$C indirect. 3D $^{15}$N-edited NOESY: HSQC Echo/Anti-echo TPPI gradient section, with Sensitivity Enhancement, without water suppression pulse in D1 (initial delay time). D8 (NOE mixing time) was set around 100–150 ms. [3D H(CCO)NH only for NF4] WATERGATE pulse scheme was used for water suppression on inverse correlation, DIPSI-2 mixing for $^{13}$C spin-lock, States-TPPI for $^1$H, $^{15}$N indirect. [3D (H)C(CO)NH only for NF4] WATERGATE pulse scheme was used for water suppression on inverse correlation, DIPSI-2 mixing for $^{13}$C spin-lock, States-TPPI for $^{13}$C, $^{15}$N indirect. [3D (H)C(C)H-TOCSY only for NF4] for aliphatic, offset placed on $^{13}$C-aliphatic center, DIPSI-3 mixing for $^{13}$C spin-lock, States-TPPI for $^{13}$C, $^{13}$C indirect; for aromatic, offset placed on $^{13}$C-aromatic center, DIPSI-3 mixing for $^{13}$C spin-lock, States-TPPI for $^{13}$C, $^{13}$C indirect.

**Structure calculation.** Several CYANA[64] calculations were performed using the ACS table, NOE peak table and dihedral angle constraints. After the CYANA calculations, several dihedral angle constraints derived from TALOS+ revealing large violations for nearly all models in the structure ensemble were eliminated. After the averaged target function of the ensemble reached <2.0 Å$^2$, refinement calculations using Amber12 were carried out for the 20 models with the lowest target functions. TALOS+ order parameter and the number of NOE distance constraints for each residue are shown in Supplementary Fig. 11.

**NMR structure validation.** R.m.s.d. values were calculated for the 20 structures overlaid to the mean coordinates for the ordered regions, automatically identified by Filt_Robot using multidimensional nonlinear scaling[47]. RDC back calculation was performed with PALES[48] using experimentally determined values of RDC. The averaged correlation between the simulated and experimental values was obtained using

the signals, except for the residues in overlapping regions in $^1$H-$^{15}$N HSQC and the residues predicted to be an order parameter of less than 0.8 by TALOS+. Detailed methods and results are described in Table 1 and the Supplementary Text. TALOS+ order parameter and the number of NOE distance constraints for each residue are shown in Supplementary Fig. 11.

### Reporting summary
Further information on research design is available in the Nature Portfolio Reporting Summary linked to this article.

## Data availability
The solution NMR structures of the eight designs have been deposited in the Protein Data Bank under accession numbers 7BPL (NF1-14), 7BPM (NF2-02), 7BQE (NF3-03), 7BQC (NF4-04), 7BPP (NF5-03), 7BQB (NF6-02), 7BPN (NF7-04) and 7BQD (NF8-01). The NMR data have been deposited in the Biological Magnetic Resonance Data Bank under accession numbers 36327 (NF1-14), 36328 (NF2-02), 36334 (NF3-03), 36332 (NF4-04), 36330 (NF5-03), 36331 (NF6-02), 36329 (NF7-04) and 36333 (NF8-01). The computational design models are presented as Supplementary Data 1. The plasmids encoding the designed sequences are available from the author upon request. ECOD database is available at http://prodata. swmed.edu/ecod. Source data are provided with this paper.

## Code availability
The codes for creating a list of all β-sheet topologies, calculating frustrations for a given β-sheet topology, calculating β-sheet topology for a given PDB file and calculating occupancy ratio for each ECOD domain entry are available at https://github.com/kogalab21/ novel_ab-fold_design.

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

## Acknowledgements

We thank the RIKEN Yokohama NMR Facility for the NMR measurements (Yokohama); the Functional Genomics Facility, NIBB Core Research Facilities (Okazaki), especially Y. Makino, for mass spectrometry analysis; and the Instrument Center (Okazaki) for HSQC NMR measurements. We also thank M. Yamamoto, N. Kobayashi and M. Kondo for support with and discussion of the experiments; K. Sakuma for discussion of the computational design; T. Kosugi for support with and discussion of the experiments, and constructive discussion on the manuscript; M. Sasai, M. Ota and D. Baker for valuable comments and discussion of the manuscript. The computation was performed using Research Center for Computational Science, Okazaki (project: 22-IMS-C188, 21-IMS-C174, 20-IMS-C157, 19-IMS-C175, 18-IMS-C155, 17-IMS-C147). The NMR structure determination was supported through the Basis for Supporting Innovative Drug Discovery and Life Science Research (BINDS) program from the Japan Agency for Medical Research and Development (AMED) under grant no. JP22am0101072. This work was supported by the Astrobiology Center Program of National Institutes of Natural Sciences (NINS) (grant no. AB291007) and the Japan Society for the Promotion of Science (JSPS) KAKENHI Grants-in-Aid for Scientific Research 15H05592 to N. Koga, 18H05420 to N. Koga, 18K06152 to N. Kobayashi and 19H03166 to G.C. and N. Koga; JST-Mirai Program (JPMJMI17A2 to N. Kobayashi); BINDS from AMED under grant no. JP20am0101111 to G.C.; and JSPS Research Fellowship (PD) 17J02339 to S.M.

## Author contributions

S.M., N. Kobayashi, R.K., G.C. and N. Koga designed the research. S.M. performed the database analysis, designed the proteins and performed the experimental characterizations except for the NMR structure determination. G.C. provided the core idea for the database analysis. N. Kobayashi, T.S. and T.N. performed the NMR structure determination experiments. N. Kobayashi performed the NMR structural analysis. S.M., N. Kobayashi., T.F., R.K., G.C. and N. Koga wrote the manuscript.

## Competing interests

The authors declare no competing interests.

## Additional information

**Extended data** is available for this paper at https://doi.org/10.1038/s41594-023-01029-0.

**Correspondence and requests for materials** should be addressed to Nobuyasu Koga.

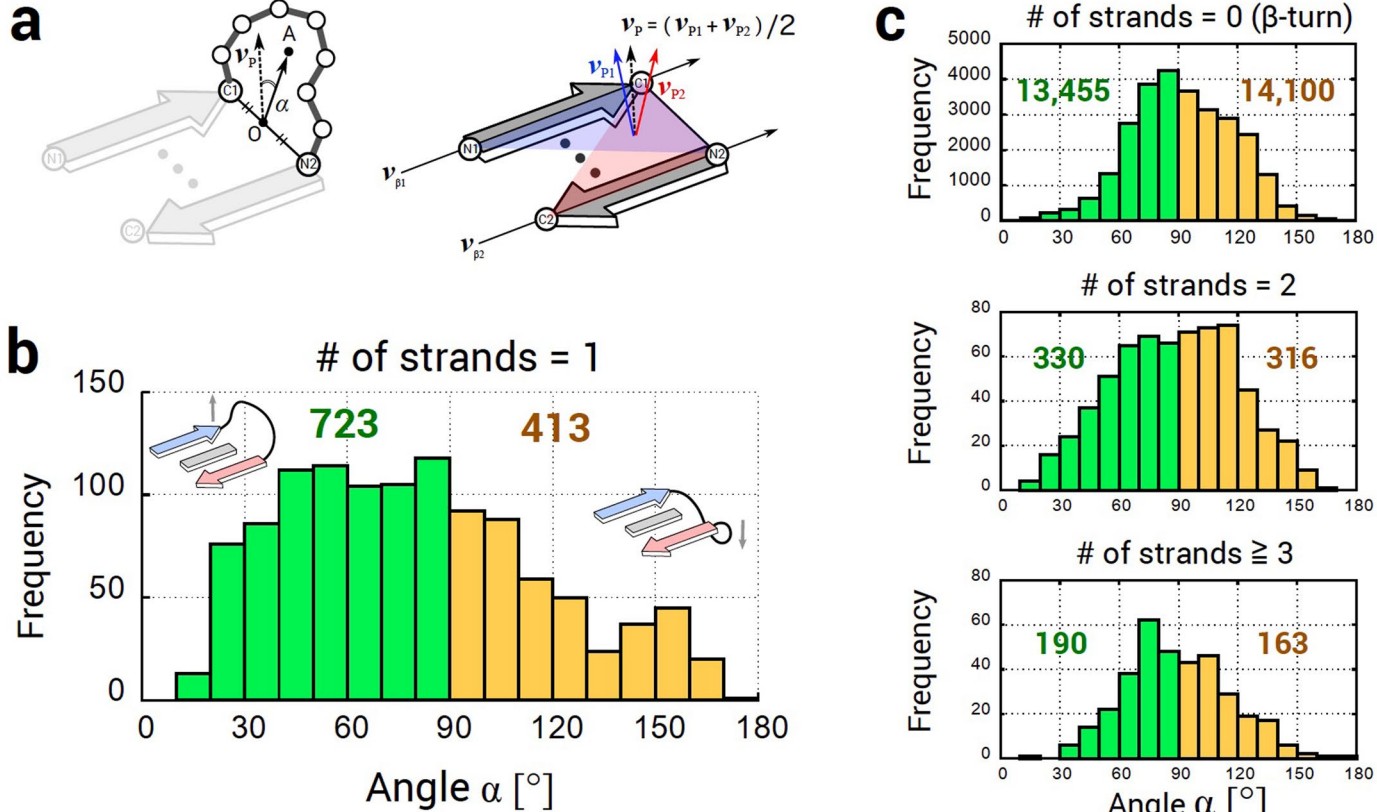

**Extended Data Fig. 1 | Bending orientation preference of anti-β-X-β motifs with a connection jump-distance number of one. a**, Left: the bending angle, α, for anti-$\beta_1$-X-$\beta_2$ motifs, identified as the angle between the β-sheet normal vector $\boldsymbol{v}_p$ and the vector from the midpoint O of the terminal β-strand backbone atoms, C1 (carbonyl carbon of the first strand) and N2 (amide nitrogen of the second strand), to the average coordinate A over the loop Cα atoms. Right: $\boldsymbol{v}_p$ calculated by averaging the normal vectors to the two planes defined by the N1-C1-N2 and C1-N2-C2 backbone atoms, respectively. **b**, Distribution of the angle α for naturally occurring protein structures with jump-distance number of one. Anti-β-X-β motifs with a bending angle < 90° are more frequently observed than those with > 90°, indicating the right-handed bending orientation preference of anti-$\beta1$-X-$\beta2$ motifs with a jump-distance number of one. This preference may arise from the intrinsic chirality and geometrical preferences of the polypeptide chain. **c**, Distributions of the angle α for naturally occurring protein structures with a jump-distance number of 0 (top), 2 (middle), and ≥3 (bottom), respectively. No bending angle preferences were observed.

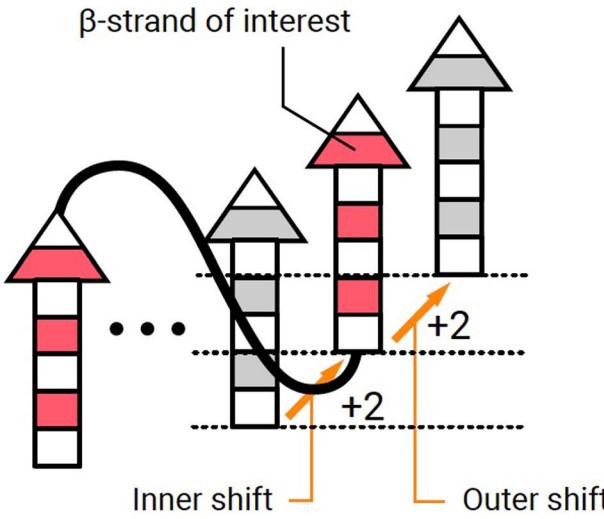

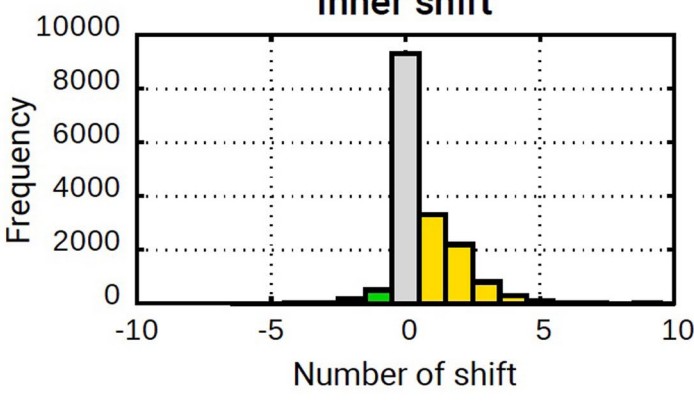

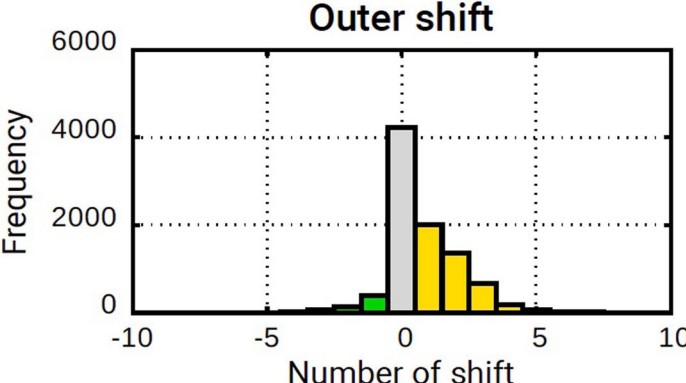

**Extended Data Fig. 2 | Register shift rule for para-β-X-β motifs[17].** Register shifts for para-β-X-β motifs were defined in the relations of the second strand (red) in the β-X-β motif with the adjacent parallelly aligned β-strands (gray): an inner register shift is when the gray β-strand is inside the para-β-X-β motif, and an outer register shift is when the gray β-strand is outside the para-β-X-β motif. Analysis of the residue offset for the inner and outer shifts for para-β-X-β motifs in naturally occurring protein structures revealed that the register shifts are mostly zero or positive; the origin of this preference is partly explained by energetic penalties of steric repulsion and buried polar atoms that emerge when unfavored register shifts occur.

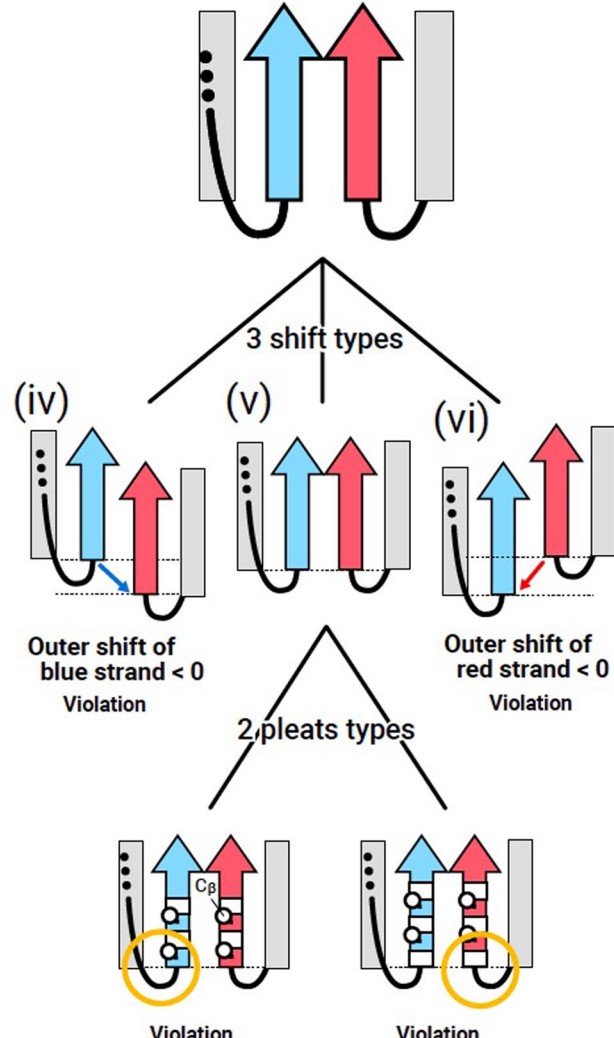

**Extended Data Fig. 3 | Origin of the connection ending rule.** Inner and outer register shift arrangements for para-para-β-X-β motifs (red and blue) violating the connection ending rule are shown on the left and right, respectively (only the second strands of the para-β-X-β motifs are shown). In the arrangements, the second strands are adjacent to each other and the connections are on the different β-sheet sides, which do not satisfy the register-shift rule[17] (Extended Data Fig. 2) or the αβ-rule[8]. In case of a register shift of non-zero [(i), (iii), (iv),

and (vi)], the β-X-β motifs violate the register shift rule (Extended Data Fig. 2). In (i) and (iv), the red strand is shifted towards the negative orientation against the blue strand; in (iii) and (vi), the blue strand is shifted towards the negative orientation against the red strand. In case of a register shift of zero [(ii) and (v)], the αβ-rule is violated: the vector from the Cα to Cβ atoms of the first strand residue in either of the β-strands points towards the X region in β-X-β motifs.

## a

Definition of twist angle μ

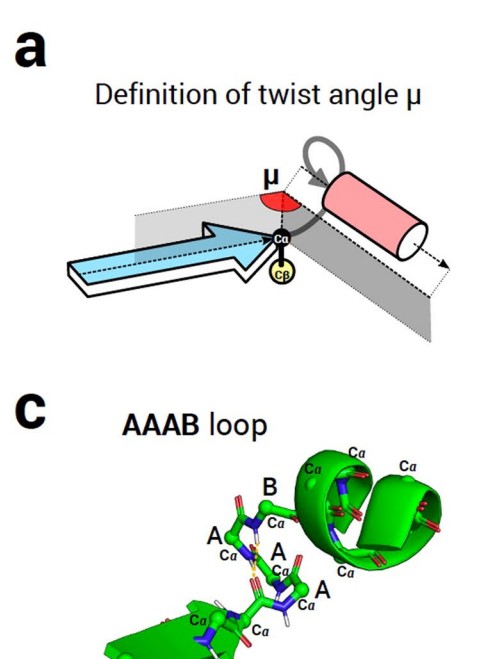

## c

AAAB loop

## b

Anti βα-loops
(90−45° < μ < 90+45°)

| Loop type | Frequency |
|-----------|-----------|
| BB | 171 |
| E | 60 |
| BAB | 45 |
| BBAAB | 23 |
| **AAAB** | 20 |
| GB | 12 |
| BBAA | 11 |
| GBB | 8 |
| AA | 6 |
| BAE | 6 |

Distribution of twist angle μ

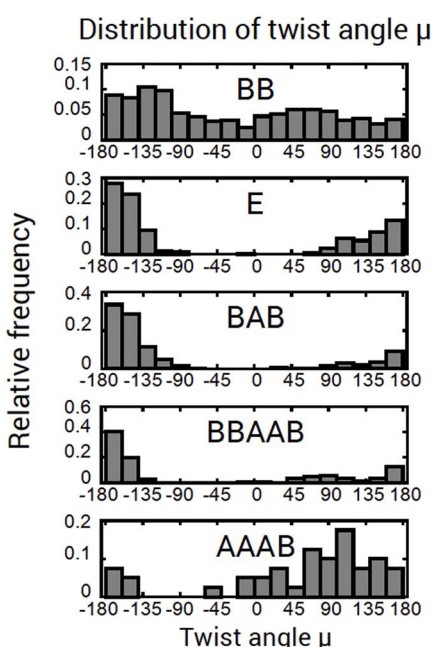

**Extended Data Fig. 4 | AAAB loop with the right twist angle for βα-units.**
**a**, The twist angle μ for βα-units of the anti-type[8] (vector from the Cα to Cβ atoms of the last strand residue points away from the helix), defined as the dihedral angle between the plane defined by the β-strand vector and the CαCβ vector of the last strand residue, and the plane defined by the same CαCβ vector and the α-helix vector (the definitions of the β-strand and α-helix vectors have been described previously[8]. **b**, Left: frequencies for ABEGO torsion patterns of loops in βα-units having a μ angle around 90° in naturally occurring protein structures. Right: distributions of the twist angle μ for each of the most frequently observed five loop types in the table on the left. The AAAB loop showing a clear peak at −90° was used in the NF7 fold design. **c**, The backbone structure of the AAAB loop.

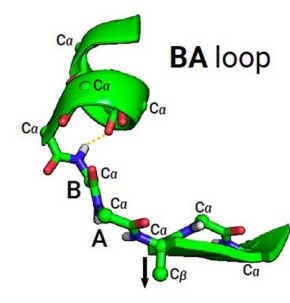

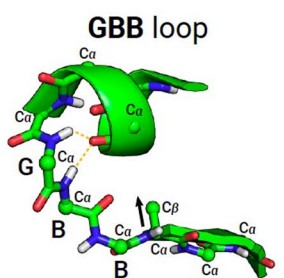

| Para-type | |
|---|---|
| loop type | Frequency |
| GB | 2,661 |
| GBA | 1,791 |
| **BA** | 550 |
| BAAB | 435 |
| **GABA** | 377 |
| GBBB | 312 |
| AGB | 261 |
| BAABA | 238 |
| ABA | 175 |
| GBBBA | 171 |

| Anti-type | |
|---|---|
| loop type | Frequency |
| **GBB** | 1,329 |
| BAA | 242 |
| BABB | 220 |
| GBAB | 165 |
| BAABB | 149 |
| AGBB | 107 |
| BBG | 94 |
| BBBG | 79 |
| ABABB | 76 |
| BAB | 73 |

**Extended Data Fig. 5 | Newly introduced loop patterns for αβ-units.**
Frequencies of ABEGO torsion patterns for the loops in αβ-units in naturally occurring proteins are shown for the para- (left) and anti-types (right) (para-type: the vector from the Cα to Cβ atoms of the first strand residue points away from the helix; anti-type: the same vector points towards the helix). The GB, GBA, and BAAB loops have been used in previous *de novo* designed proteins[8,9]. The BA and GABA loops for the para-type and the GBB loop for the anti-type were newly introduced in this study.

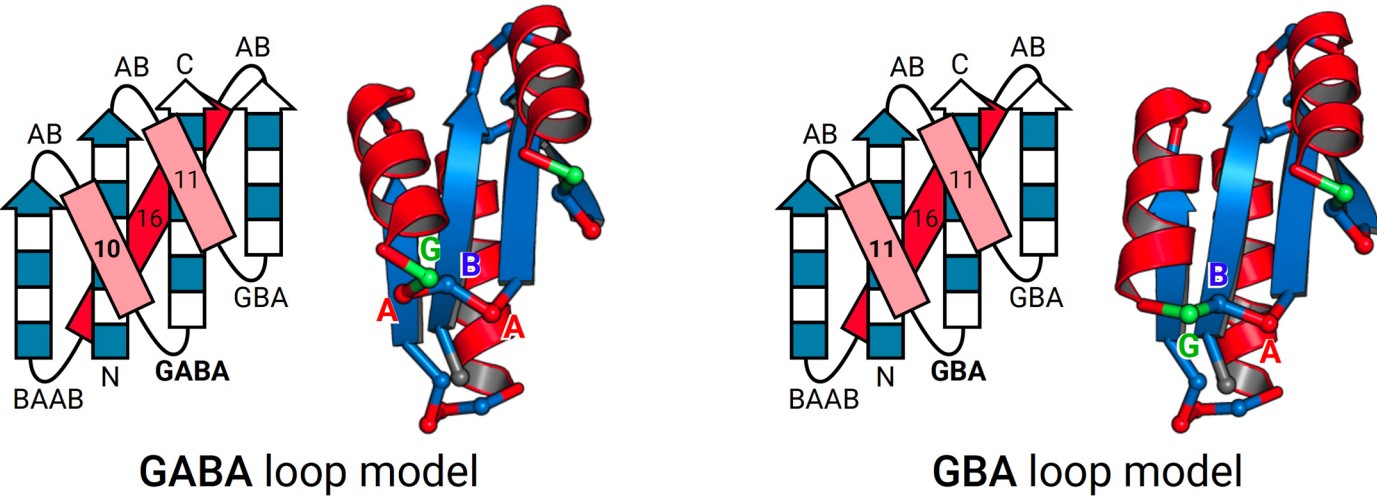

**Extended Data Fig. 6 | Two backbone blueprints used for the design of the target NF8.** The torsion patterns immediately before the last strand are different.

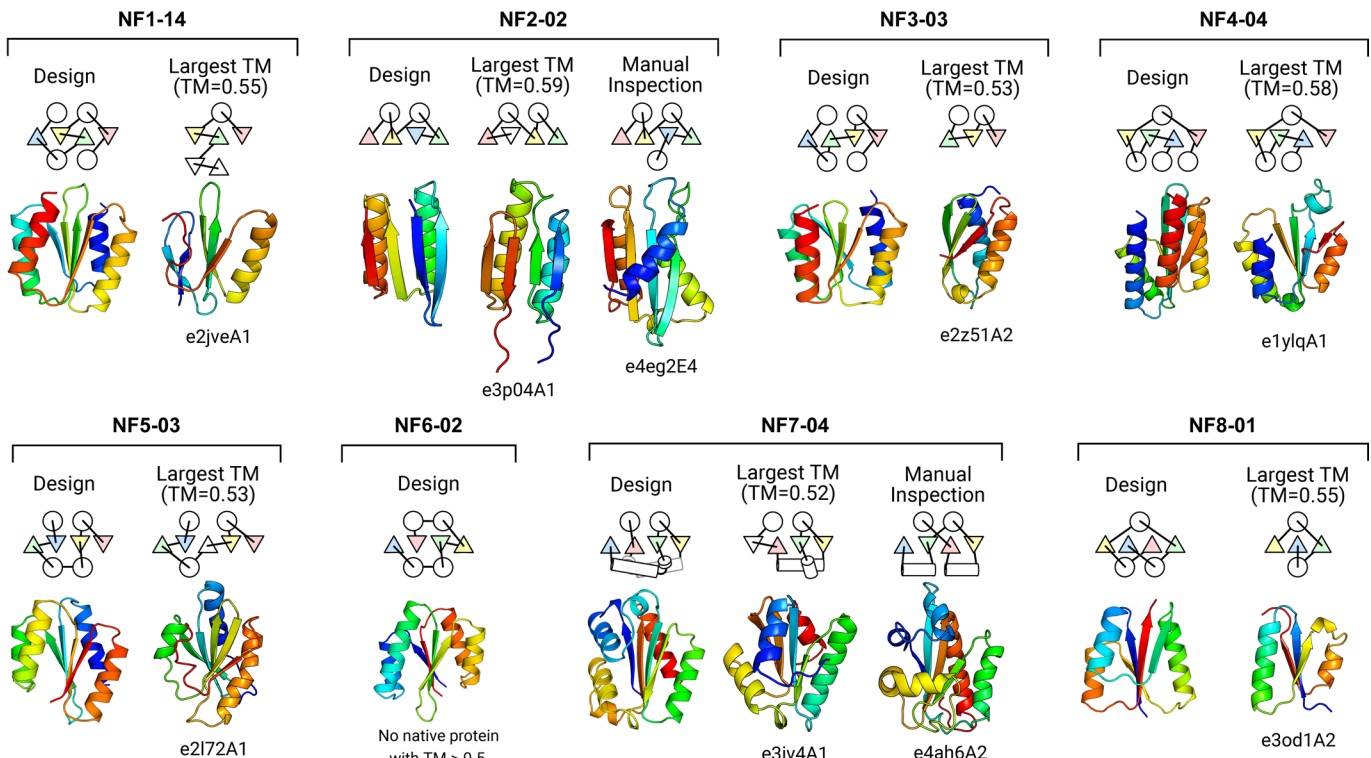

**Extended Data Fig. 7 | Structure search for naturally occurring proteins similar to the designs in terms of entire structures.** For each designed structure, similar domain structures were searched against the ECOD domain dataset[48] (99% sequence non-redundant set) using the two different TM-score[25] -based structure alignment methods, TM-align[25] and MICAN[26,27] (sequential mode) (Different from TM-align[25], MICAN[26,27] superimposes structures using secondary-structure-weighted TM-score[25]). We collected all domains with a TM-score > 0.5 compared to each target structure and inspected them manually using the TOPS diagram[28]. The domain with the largest TM-score for each target except for NF6-02 (there is no domain with a TM-score > 0.5) and the domain similar to each of the NF2 and NF7 designs, found by the manual inspection, were shown in each panel together with ECOD ID. No similar naturally occurring protein structures were found for the designs, except for the NF2 and NF4 designs.

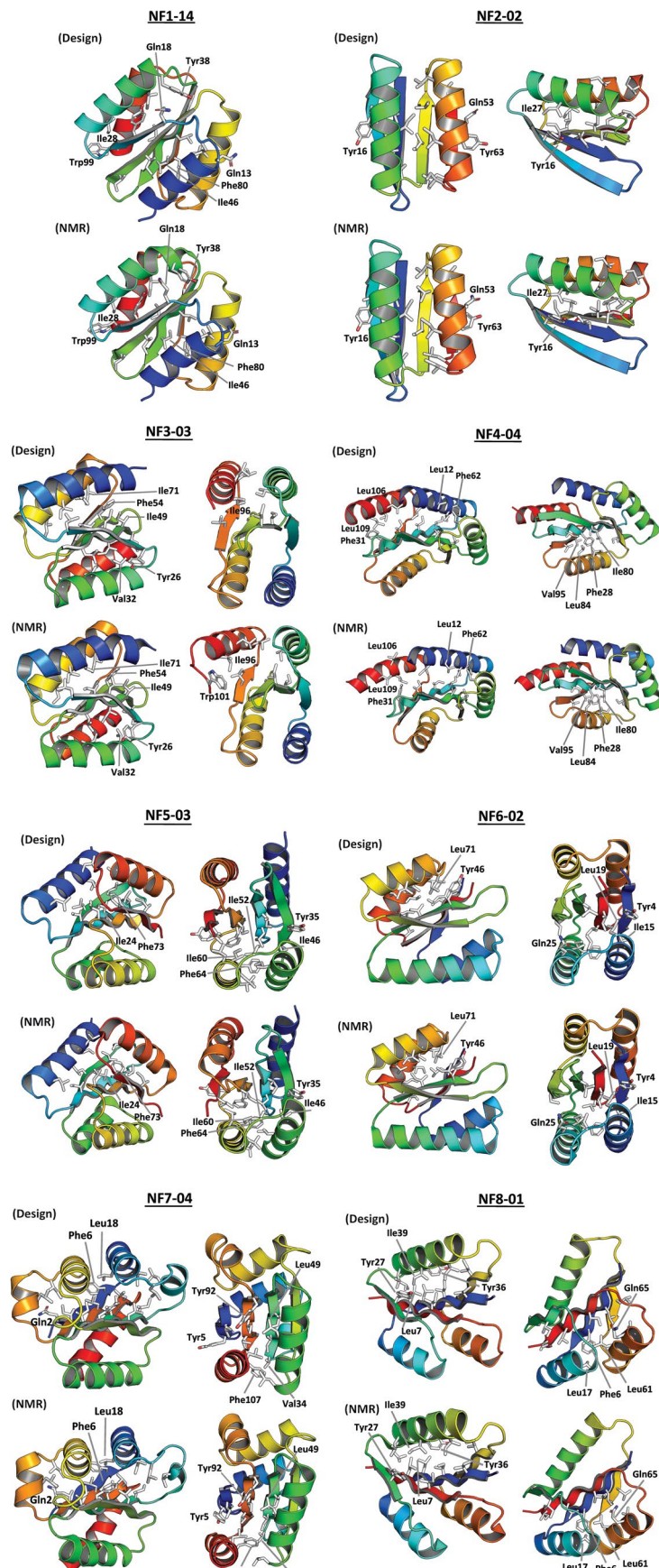

**Extended Data Fig. 8 | Comparison of core packing between design models and NMR structures.** Hydrophobic residues in core, mainly for Leu, Ile, Phe, Tyr, and Trp, are shown in stick. For the residues with amino-acid type and residue number, detail descriptions in terms of HSQC spectra are provided in the Supplementary text.

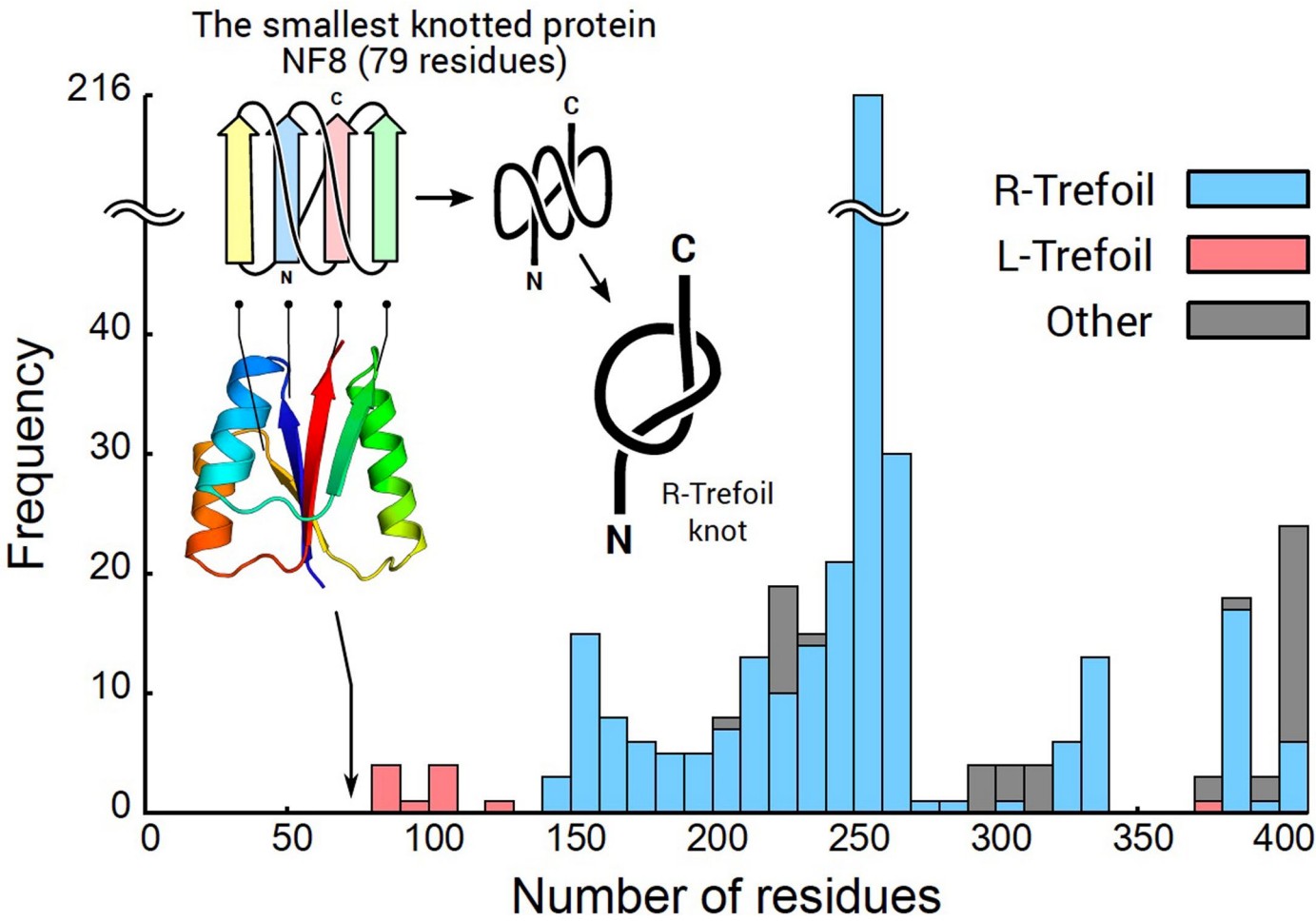

**Extended Data Fig. 9 | Smallest knotted protein designed, NF8.** The stacked histogram represents the number of naturally occurring knot proteins in the PDB, depending on the chain length (original annotation data were obtained from the KnotProt database[65]. Blue, red, and gray bars represent right-handed trefoil knot (R-Trefoil), left-handed trefoil knot (L-Trefoil), and other knot types (Other), respectively. The design NF8 with the R-Trefoil knot, indicated by an arrow, is characterized as the smallest knotted protein with 79 residues. Note that this is an exceptional case for R-Trefoil knot structures; the minimal size observed in nature is approximately 140 residues (the smallest L-Trefoil structure has 82 residues).

**Extended Data Table 1 | Summary of experimental results for the designed proteins**

| | #Designs tested | Expressed[1] | Soluble[1] | αβ-protein CD spectrum (20 °C) | Monomeric[2] | Well-resolved HSQC[3] | Success (rate %) |
|---|---|---|---|---|---|---|---|
| **NF1** | 16 | 16 | 14 | 14 | 6 | 4 | 4 (25) |
| **NF2** | 4 | 4 | 4 | 4 | 4 | 2 | 2 (50) |
| **NF3** | 4 | 4 | 3 | 3 | 3 | 2 | 2 (50) |
| **NF4** | 6 | 6 | 6 | 6 | 5 | 4 | 4 (67) |
| **NF5** | 6 | 6 | 6 | 6 | 2 | 2 | 2 (33) |
| **NF6** | 6 | 6 | 6 | 6 | 5 | 5 | 5 (83) |
| **NF7** | 6 | 6 | 6 | 6 | 3 | 3 | 3 (50) |
| **NF8** | 12 | 12 | 11 | 11 | 6 | 6 | 6 (50) |

The second column shows the number of designs experimentally tested for the fold in the first column. The following columns show the number of designs that satisfy the experimental characterizations, which were performed sequentially from the left to the right. Successful designs are defined as those that satisfy all criteria. Details of the results are shown in Supplementary Tables 10–17. [1] Expression and solubility were assessed by sodium dodecyl sulfate polyacrylamide gel electrophoresis and mass spectrometry. [2] Size-exclusion chromatography combined with multi-angle light scattering was used to determine the oligomerization state. The number of designs in which the main peak of the absorbance at 280 nm corresponds to the monomeric state was counted. [3] $^{1}$H-$^{15}$N heteronuclear single quantum coherence (HSQC) spectra were collected.

# Reporting Summary

## Statistics

For all statistical analyses, confirm that the following items are present in the figure legend, table legend, main text, or Methods section.

| n/a | Confirmed | |
|---|---|---|
| ☐ | ☒ | The exact sample size (*n*) for each experimental group/condition, given as a discrete number and unit of measurement |
| ☐ | ☒ | A statement on whether measurements were taken from distinct samples or whether the same sample was measured repeatedly |
| ☒ | ☐ | The statistical test(s) used AND whether they are one- or two-sided *Only common tests should be described solely by name; describe more complex techniques in the Methods section.* |
| ☒ | ☐ | A description of all covariates tested |
| ☒ | ☐ | A description of any assumptions or corrections, such as tests of normality and adjustment for multiple comparisons |
| ☐ | ☒ | A full description of the statistical parameters including central tendency (e.g. means) or other basic estimates (e.g. regression coefficient) AND variation (e.g. standard deviation) or associated estimates of uncertainty (e.g. confidence intervals) |
| ☒ | ☐ | For null hypothesis testing, the test statistic (e.g. *F*, *t*, *r*) with confidence intervals, effect sizes, degrees of freedom and *P* value noted *Give P values as exact values whenever suitable.* |
| ☒ | ☐ | For Bayesian analysis, information on the choice of priors and Markov chain Monte Carlo settings |
| ☒ | ☐ | For hierarchical and complex designs, identification of the appropriate level for tests and full reporting of outcomes |
| ☒ | ☐ | Estimates of effect sizes (e.g. Cohen's *d*, Pearson's *r*), indicating how they were calculated |

*Our web collection on statistics for biologists contains articles on many of the points above.*

## Software and code

Policy information about availability of computer code

| Data collection | Rosetta software suite 3 was used for protein design and folding calculations. JASCO SpectraManager software v2 was used for CD. |
|---|---|
| Data analysis | The codes for creating a list of all β-sheet topologies, calculating frustrations for a given β-sheet topology, calculating β-sheet topology for a given pdb file, and calculating occupancy ratio for each ECOD domain entry will be available at https://github.com/kogalab23/novel_ab-fold_design. The database analysis for the investigation of similar naturally occurring protein structures was performed by MICAN (2018.04.05) and TM-align (20120126). For all obtained domain structures from ECOD database (20180423-(develop210)-F99), structure refinements were performed using ModRefiner, and the secondary structures were assigned using STRIDE. SEC-MALS data were analyzed by ASTRA software 6.1.2. HSQC data were analyzed by Delta 5.0.4 NMR software. All NMR structure analyses were done as described in the methods section with the following programs: MagRO-NMRViewJ (updated version of Kujira),Filt_Robot, TALOS+ 2017, FLYA 3.98.5, CYANA 3.98.5, Amber12, and PALES 2.1. |

For manuscripts utilizing custom algorithms or software that are central to the research but not yet described in published literature, software must be made available to editors and reviewers. We strongly encourage code deposition in a community repository (e.g. GitHub). See the Nature Portfolio guidelines for submitting code & software for further information.

## Data

Policy information about availability of data

All manuscripts must include a data availability statement. This statement should provide the following information, where applicable:

- Accession codes, unique identifiers, or web links for publicly available datasets
- A description of any restrictions on data availability
- For clinical datasets or third party data, please ensure that the statement adheres to our policy

The solution NMR structures have been deposited in the wwPDB as PDB 7BPL (NF1-14), 7BPM (NF2-02), 7BQE (NF3-03), 7BQC (NF4-04), 7BPP (NF5-03), 7BQB (NF6-02), 7BPN (NF7-04), and 7BQD (NF8-01). The NMR data have been deposited in the BMRB under the accession numbers 36327 (NF1-14), 36328 (NF2-02), 36334 (NF3-03), 36332 (NF4-04), 36330 (NF5-03), 36331 (NF6-02), 36329 (NF7-04), and 36333 (NF8-01). The computational design models are presented as Supplementary Data 1. The plasmids encoding the designed sequences are available from the author upon request. ECOD database is available at http://prodata.swmed.edu/ecod. A dataset obtained from cullpdb database will be available at https://github.com/kogalab23/novel_ab-fold_design.

# Field-specific reporting

Please select the one below that is the best fit for your research. If you are not sure, read the appropriate sections before making your selection.

☒ Life sciences          ☐ Behavioural & social sciences          ☐ Ecological, evolutionary & environmental sciences

For a reference copy of the document with all sections, see nature.com/documents/nr-reporting-summary-flat.pdf

# Life sciences study design

All studies must disclose on these points even when the disclosure is negative.

| | |
|---|---|
| Sample size | Computational designs that passed selection criteria were experimentally tested. Based on the previously reported success rate of de novo designed proteins of similar size (N. Koga et al. Nature, 2012; Y.-R. Lin et al., PNAS, 2015), we estimated the number of designs we should test in order to be successful. |
| Data exclusions | No data were excluded |
| Replication | The representative designs for NMR structure determination were purified, verified by SDS-PAGE, mass spectrometry, and HSQC measurements more than once. All attempts at replication were successful. |
| Randomization | Randomization is not relevant to our study. This is an observational study, which does not involve evaluation of conditional effects. |
| Blinding | Blinding is not relevant to our study. Keeping track of the identity of each designed protein was necessary for characterizing biophysical properties and solving the structure. |

# Reporting for specific materials, systems and methods

We require information from authors about some types of materials, experimental systems and methods used in many studies. Here, indicate whether each material, system or method listed is relevant to your study. If you are not sure if a list item applies to your research, read the appropriate section before selecting a response.

### Materials & experimental systems

| n/a | Involved in the study |
|---|---|
| ☒ | ☐ Antibodies |
| ☒ | ☐ Eukaryotic cell lines |
| ☒ | ☐ Palaeontology and archaeology |
| ☒ | ☐ Animals and other organisms |
| ☒ | ☐ Human research participants |
| ☒ | ☐ Clinical data |
| ☒ | ☐ Dual use research of concern |

### Methods

| n/a | Involved in the study |
|---|---|
| ☒ | ☐ ChIP-seq |
| ☒ | ☐ Flow cytometry |
| ☒ | ☐ MRI-based neuroimaging |

