## [Peer Review File · Nature Structural & Molecular Biology]

Peer Review Information

Manuscript Title: Exploration of novel $\alpha\beta$ -protein folds through de novo design

Corresponding author name(s): Nobuyasu Koga

Reviewer Comments & Decisions:

Decision Letter, initial version:
--

Message: 7th Apr 2022

Dear Dr. Koga,

Thank you again for submitting your manuscript "Exploration of novel $\alpha\beta$ -protein folds through de novo design". I apologize for the delay in responding, which resulted from the difficulty in obtaining suitable referee reports. Nevertheless, we now have comments (below) from the 3 reviewers who evaluated your paper. In light of those reports, we remain interested in your study and would like to see your response to the comments of the referees, in the form of a revised manuscript.

You will see that specifically reviewer #3 questions whether the NMR structures may be over-constrained. Please be sure to address/respond to all concerns of the referees in full in a point-by-point response and highlight all changes in the revised manuscript text file. If you have comments that are intended for editors only, please include those in a separate cover letter.

We expect to see your revised manuscript within 6 weeks. If you cannot send it within this time, please contact us to discuss an extension; we would still consider your revision, provided that no similar work has been accepted for publication at NSMB or published elsewhere.

Reporting Summary:

When submitting the revised version of your manuscript, please pay close attention to our [href="https://www.nature.com/nature-research/editorial-policies/image-integrity">Digital Image Integrity Guidelines. and to the following points below:](https://www.nature.com/nature-research/editorial-policies/image-integrity)

Please note that all key data shown in the main figures as cropped gels or blots should be presented in uncropped form, with molecular weight markers. These data can be aggregated into a single supplementary figure item. While these data can be displayed in a relatively informal style, they must refer back to the relevant figures. These data should be submitted with the final revision, as source data, prior to acceptance, but you may want to start putting it together at this point.

Data availability: this journal strongly supports public availability of data. All data used in accepted papers should be available via a public data repository, or alternatively, as Supplementary Information. If data can only be shared on request, please explain why in your Data Availability Statement, and also in the correspondence with your editor. Please note that for some data types, deposition in a public repository is mandatory - more information on our data deposition policies and available repositories can be found below: <https://www.nature.com/nature-research/editorial-policies/reporting-standards#availability->

of-data

[Redacted]

Sincerely,
Sara

Sara Osman, Ph.D.
Associate Editor
Nature Structural & Molecular Biology

Referee expertise:

Referee #1: Protein design, evolution

Referee #2: Protein design, computational biology

Referee #3: NMR

Reviewers' Comments:

Reviewer #1:

Remarks to the Author:

A central question in structural biology asks whether the protein folds that occur in life on earth represent all the folds that are possible. Or merely a small subset of possible folds. In recent years, this question has moved to the forefront as new structure determinations almost always reveal folds that have already been seen many times before. Is that all there is? Or are new folds possible?

When asking what is possible, perhaps the best way to answer the question leads to De Novo design. With respect to novel folds, the design-to-asses-what-is-possible approach was pioneered in 2003 by Kuhlman et al. (Ref. 13.) The current manuscript takes this approach a giant step forward. Instead of merely asking about one possible fold/topology (as was done in Ref. 13), Minami et al. aim to assess all possible topologies for alpha/beta protein folds.

The authors initiate their study by enumerating all possible topologies and then devising rules about which connections between units of secondary structure are favored or disfavored. This approach is based on a deep dive into the geometry of protein structures. Geometrical approaches to protein structure were pioneered in the 1970s by Jane Richardson. Koga has emerged as the current leader in this field, using a combination of intuition and computational tools to describe the rules of protein topology.

After developing their rules for which topologies are favored or disfavored, Minami et al. divide all the imaginable folds into two groups: Those that are frustrated (i.e. cannot simultaneously satisfy all the rules) and those that are non-frustrated (satisfy all the rules.) Whether or not this latter group is truly non-frustrated (or just seems that way from their theoretical treatment) can then be tested by going to the laboratory and designing novel amino acid sequences to fold into these new topologies.

Minami et al. used computational approaches, based primarily on Rosetta, to design a series of proteins that fold into eight different topologies, which according to their rules, are non-frustrated and therefore possible, but which have not been observed (or observed very rarely) in the PDB.

Experimental characterization of their novel proteins revealed their approach is spectacularly successful. All the new topologies they attempted were successfully designed. The experimentally determined structures were nearly identical to the designed targets, and the proteins were very stable -- well beyond most natural proteins.

Overall, this study represents a seminal step forward in the field. I say this for two reasons: First, the authors demonstrate explicitly and experimentally that novel structures beyond those seen in nature are possible, and can be very stable. Second, the current manuscript represents a dramatic step forward in the field of de novo protein design. While Kuhlman et al. showed in 2003 that one novel fold can be designed, the current manuscript suggests that irrespective of whether a fold exists in nature, all non-frustrated topologies can be designed and produced in the laboratory !

The text itself is extremely well written. It is a pleasure to read. The figures are superb. They convey an enormous amount of complex information, yet through creative choice of formatting, symbols, and colors, the authors make this complex information accessible to any reader who takes the time to study the figures.

Prior to publication, however, there are a few things that must be fixed. Some of these are very important, but they're all easy to fix.

Lines 89-95: This section ends with a sentence stating "These ... led to the connection overlap rule (Fig. 2b)." However, the text doesn't tell the reader what this rule is. The figure is reasonably clear demonstrating that S (same side) is typically less favored than D (different side.) But please, this should be explained in the text itself.

Line 97-105: Here I have a similar concern. The text states "These two preferences led to the connection ending rule." Okay, but what is the connection ending rule? Please tell us.

Line 133: The subtitle claims the study designs all the predicted novel four stranded alpha-beta folds. I think it would be more correct to say all the non-frustrated folds?

Line 142: The text mentions ABEGO, but never tells us what this is about. This referee understood this terminology from reading earlier papers in this field. However, many readers will not have that background. Therefore, the manuscript must explain this nomenclature.

Line 169: "The proteins were expressed in E. coli and purified using a Ni-NTA affinity column." Something is missing here. Please revise to the following: "The proteins were expressed in E. coli with C terminal His tags and purified using a Ni-NTA affinity column."

Line 211. Minor typo: "these novel folds have not observed in nature." Should be "these novel foals have not been observed in nature."

Lines 211-217. The discussion asks why novel folds have not been observed in nature. I'd like to add a suggestion: All life on earth descended from common ancestry. Therefore, life on earth is biased by this ancestral relationship. (In some ways, all protein sequences are "cousins.") Protein design allows one to explore beyond this family of cousins by sampling sequences that are novel and did not arise from common ancestry.

Figure 2b2 top line. These two pictures look the same. The only difference is the color of the turn. However, the legend doesn't explain the colors. Therefore, I did not see any difference.

Figure 2c top line. As above, the only difference was the color. But what does the color mean?

Figure 3D. Perhaps I am misunderstanding this, but I think there are some mistakes here. The first column is titled 1234, and every picture in that column orders the strands 1234. Likewise, the second column is 2134 and again they all go 2134. But then things get strange. The third column lists 1423, but the pictures seem to go 1342. The next column is titled 4123 but that's not how the pictures go. Another column lists 1342 but the pictures seem to go 1423. Also consider new protein 8. It's in a column labeled 3142. However, the figure does not do that! Perhaps some of the columns are mislabeled? Or perhaps I am misunderstanding something. Anyway, there seem to be many errors in this figure and something needs to be clarified

In the big picture though, this is a fantastic paper. It needs a few corrections but those should be easy to fix.

Reviewer #2:

Remarks to the Author:

In this most interesting manuscript, the authors show that they can design proteins with new topologies and do so on a large scale. This result is highly significant, because it essentially answers the question whether nature has probed all possible topologies among naturally occurring proteins. Apparently, many folds not found in nature are physically possible, and sequences that fold into them exist. This is a powerful message and it is delivered very well through this work.

The work is solid, well-planned, meticulously executed and lucidly presented. The structures of designed proteins were determined experimentally. To simplify their task, the authors discredited some topologies by establishing a number of rules. These rules do not allow topologies that would be energetically unfavorable in most obvious ways, and therefore would be most difficult to design sequences into. However, the question arises, if sequences can still be designed even for these most challenging topologies. I would not be surprised that it would be possible, and I hope the authors would explore such designs when their design software improves.

My only concern is about the search of protein structures for given topologies. The authors claim that their designs represent new topologies, but this claim is not very well supported in the current version of the manuscript. In fact, the suppl. Fig. 5 shows the opposite: e.g. NF2 is not a new fold at all, because "manual inspection" reveals the protein with exactly the same topology. And for the algorithm, they rely on TM-align, which is not an appropriate method for the task. TM-align best hits are misleading because TM-align simply maximizes spatial proximity of residues in superimposed structures and does not explicitly constrain the topology. Therefore, it would be very nice if the authors use some tools that find proteins that contain given topologies as their substructures. Then, they should find the "manual inspection" hit for NF2, and possibly some more for other "new folds".

This comment does not diminish the astonishing success of the authors in their design of uncommon topologies, but it would be interesting to know which of these topologies can still be found among known structures, and which ones are truly new. BTW, I would also consider NF4 to be present in PDB: even TM-align finds the right hit (missing the last helix, but it is a minor detail).

Finally, just out of curiosity, would alphafold predict the structures of these new fold proteins correctly? And are these designed folds present in the alphafold database of predicted structures?

Reviewer #3:

Remarks to the Author:

The NMR experiments for this manuscript are generally well described. However, there are a number of points that should be addressed.

1. More detail on assignment of side chain resonances should be provided in the Materials section. Specifically, what pulse sequences were used for aliphatic and aromatic assignments?
2. How does the core packing compare between the designed and NMR structures? What are the RMSDs for all heavy atoms? (not just the backbone). Descriptions of core packing of NMR structures in the supplement would be easier to follow if there were accompanying figures showing the relevant regions.
3. How do the TALOS derived order parameters compare with the RMSDs in the structures? It would be useful to map the order parameters onto the structures. Overall, the NMR ensembles seem to be very tightly constrained, even in loop regions, which is a bit surprising. Do these highly constrained loops also have very high order parameters? If they don't, then the structures are probably over constrained.
4. Another aspect that is somewhat concerning, and may be related to point 3, is that the thermal denaturation profiles for NF4,5, and 6 do not seem to be very cooperative. Can the authors comment? It does not seem that this would be consistent with the very highly constrained structures that are being generated. Linewidths for NF4, in particular, seem to be exchange broadened compared with some of the other proteins, suggesting the possibility of a more flexible structure than what has been

Author Rebuttal to Initial comments

To Reviewer #1:

We sincerely appreciate your comments and suggestions. We revised our manuscript according to them. Point-by-point responses are described in the following.

1-1) Lines 89-95: This section ends with a sentence stating “These ... led to the connection overlap rule (Fig. 2b).” However, the text doesn't tell the reader what this rule is. The figure is reasonably clear demonstrating that S (same side) is typically less favored than D (different side.) But please, this should be explained in the text itself.

Thank you for the suggestion. We responded to this together with the next suggestion.

1-2) Line 97-105: Here I have a similar concern. The text states “These two preferences led to the connection ending rule.” Okay, but what is the connection ending rule? Please tell us.

We had described each rule at the beginning of each paragraph, but we realized that it was not clear, as you pointed out. To clarify this, we enclosed what the rule is with double quotation mark for all three rules, and revised the rules with more explanations (Lines: 85-87, 92-94, 101-103).

2) Line 133: The subtitle claims the study designs all the predicted novel four stranded alpha-beta folds. I think it would be more correct to say all the non-frustrated folds?

Thank you for the suggestion. We added the word “non-frustrated” into the subtitle of the section, in which the rules were applied to identify frustration-free novel $\alpha\beta$ -folds (i.e. the section immediately before the section you mentioned; Line 110).

3) Line 142: The text mentions ABEGO, but never tells us what this is about. This referee understood this terminology from reading earlier papers in this field. However, many readers will not have that background. Therefore, the manuscript must explain this nomenclature.

Thank you for pointing this out. We added the explanation and references about ABEGO (Lines: 150-151).

4) Line 169: “The proteins were expressed in E. coli and purified using a Ni-NTA affinity column.” Something is missing here. Please revise to the following: “The proteins were expressed in E. coli with C terminal His tags and purified using a Ni-NTA affinity column.”

Thank you for pointing this out. We modified it (Line: 181).

5) Line 211. Minor typo: “these novel folds have not observed in nature.” Should be “these novel foals have not been observed in nature.”

We modified it (Line 228). Thank you.

6) Lines 211–217. The discussion asks why novel folds have not been observed in nature. I’d like to add a suggestion: All life on earth descended from common ancestry. Therefore, life on earth is biased by this ancestral relationship. (In some ways, all protein sequences are “cousins.”) Protein design allows one to explore beyond this family of cousins by sampling sequences that are novel and did not arise from common ancestry.

Thank you for the suggestion. We added the description into the first possible reason why novel folds have not been observed in nature (Lines: 228-231).

7-1) Figure 2b2 top line. These two pictures look the same. The only difference is the color of the turn. However, the legend doesn’t explain the colors. Therefore, I did not see any difference.

Thank you for the comment. We responded to this together with the next comment.

7-2) Figure 2c top line. As above, the only difference was the color. But what does the color mean?

The blue and red colored motifs indicate two different β -X- β motifs. We added the explanation into the Fig. 2 legend (Lines: 441-442).

8) Figure 3D. Perhaps I am misunderstanding this, but I think there are some mistakes here. The first column is titled 1234, and every picture in that column orders the strands 1234. Likewise, the second column is 2134 and again they all go 2134.

But then things get strange. The third column lists 1423, but the pictures seem to go 1342. The next column is titled 4123 but that's not how the pictures go. Another column lists 1342 but the pictures seem to go 1423. Also consider new protein 8. It's in a column labeled 3142. However, the figure does not do that! Perhaps some of the columns are mislabeled? Or perhaps I am misunderstanding something.

Anyway, there seem to be many errors in this figure and something needs to be clarified

Thank you for the comments. The notion for β -sheet topologies represents the order of β -strands, numbered along the linear sequence, in a β -sheet from left to right, not the order of the position of β -strands in a β -sheet. For example, 1342 indicates that the 1st, 3rd, 4th, and 2nd β -strands are aligned from left to right in a β -sheet. The notion can be found in the SCOP structure database, for example. We added the explanation in the Fig. 3 legend (Lines: 459-460).

In the big picture though, this is a fantastic paper. It needs a few corrections but those should be easy to fix.

Thank you very much !

To Reviewer #2:

We sincerely appreciate your comments and suggestions. We revised our manuscript according to them. Point-by-point responses are described in the following.

1) The work is solid, well-planned, meticulously executed and lucidly presented. The structures of designed proteins were determined experimentally. To simplify their task, the authors discredited some topologies by establishing a number of rules. These rules do not allow topologies that would be energetically unfavorable in most obvious ways, and therefore would be most difficult to design sequences into. However, the question arises, if sequences can still be designed even for these most challenging topologies. I would not be surprised that it would be possible, and I hope the authors would explore such designs when their design software improves.

Thank you very much for the encouraging comments. We completely agree with you !

2) My only concern is about the search of protein structures for given topologies. The authors claim that their designs represent new topologies, but this claim is not very well supported in the current version of the manuscript. In fact, the suppl. Fig. 5 shows the opposite: e.g. NF2 is not a new fold at all, because "manual inspection" reveals the protein with exactly the same topology. And for the algorithm, they rely on TM-align, which is not an appropriate method for the task. TM-align best hits are misleading because TM-align simply maximizes spatial proximity of residues in superimposed structures and does not explicitly constrain the topology. Therefore, it would be very nice if the authors use some tools that find proteins that contain given topologies as their substructures. Then, they should find the "manual inspection" hit for NF2, and possibly some more for other "new folds".

This comment does not diminish the astonishing success of the authors in their design of uncommon topologies, but it would be interesting to know which of these topologies can still be found among known structures, and which ones are truly new. BTW, I would also consider NF4 to be present in PDB: even TM-align finds the right hit (missing the last helix, but it is a minor detail).

Thank you for the comments. We realized that the definition for novel folds in our manuscript was not clear. We agree with you that the $\alpha\beta$ -folds of NF2 and NF4 are not completely new ones. In this study, we defined $\alpha\beta$ -folds based on their β -sheet topologies (i.e. the number, orientation, and order of constituent β -strands). As shown in Fig. 3, the β -sheet topologies of NF1-4 were found in nature, whereas the β -sheet topologies of NF5-8 were found to be truly new. However, the observation frequencies of NF1-4 are very low. Therefore, we regarded not only the β -sheet topologies of NF5-8, which have not been observed in nature, but also those of NF1-4 as novel because of the evolutionary instability. The structure search performed in Extended Data Fig. 5 was aimed at finding naturally occurring protein structures similar to the designs in terms of the entire backbone-structure level. We used two different TM-score-based structure alignment methods, TM-align and MICAN; the latter superimposes structures weighted on the arrangements (spatial positions and orientations) of secondary structure elements. For the domain with the largest TM-score and similar domains found by manual inspection, the domain IDs were shown. We added these explanations in the main text (Lines: 129-135, 144-145, 163-168, 533-536) and revised the description in the Extended Data Fig. 5.

3) Finally, just out of curiosity, would alphafold predict the structures of these new fold proteins correctly? And are these designed folds present in the alphafold database of predicted structures?

Thank you for the interesting comments. We performed the structure prediction by Alphafold for all eight designs, with the template database up to 5/14/2009 (This is for removing most de novo designed structures, all but Top7). The four designs (NF2, 4, 5, and 6) were correctly predicted for

all five predicted models, while the other four (NF1, 3, 7, and 8), including the one with the knotted fold, were correctly predicted for some but not all five models. We added this result in the discussion (Lines: 238-244). For the second comment, we are currently working on this as the next project. We will describe the findings in the next paper.

To Reviewer #3:

We sincerely appreciate your comments and suggestions. We revised our manuscript according to them. Point-by-point responses are described in the following.

The NMR experiments for this manuscript are generally well described. However, there are a number of points that should be addressed.

1. More detail on assignment of side chain resonances should be provided in the Materials section.

Specifically, what pulse sequences were used for aliphatic and aromatic assignments?

Thank you very much. Following your suggestion, we provided the details in the Methods section (p. 34 lines 708-725).

2. How does the core packing compare between the designed and NMR structures? What are the RMSDs for all heavy atoms? (not just the backbone). Descriptions of core packing of NMR structures in the supplement would be easier to follow if there were accompanying figures showing the relevant regions.

Thank you for pointing this out. We presented the comparison of the core packing between the designed and NMR structures in Extended Data Fig. 10 and added a description in the section “Experimental characterisation of novel four-stranded $\alpha\beta$ -folds” in the Results (Lines: 197-199). We showed the RMSD values for all heavy atoms in Supplementary Table 9.

3. How do the TALOS derived order parameters compare with the RMSDs in the structures? It would be useful to map the order parameters onto the structures. Overall, the NMR ensembles seem to be very tightly constrained, even in loop regions, which is a bit surprising. Do these highly constrained loops also have very high order parameters? If they don't, then the structures are probably over constrained.

Thank you for the comment. Following your comment, we plotted the TALOS derived order parameter for each residue for each designed structure in Supplementary Fig. 9. As shown in the figure, high order parameters of more than 0.6 were observed not only for the residues in secondary structures but also for those in loops. Moreover, we plotted the number of NOE distance constraints for each residue in the figure; most residues have enough long- or short- range distance constraints. Therefore, the structural convergence of the NMR models we determined should not be a problem.

4. Another aspect that is somewhat concerning, and may be related to point 3, is that the thermal denaturation profiles for NF4,5, and 6 do not seem to be very cooperative. Can the authors comment? It does not seem that this would be consistent with the very highly constrained structures that are being generated. Linewidths for NF4, in particular, seem to be exchange broadened compared with some of the other proteins, suggesting the possibility of a more flexible structure than what has been determined.

Thank you for the comment. As you pointed out, the thermal denaturation profiles for the designed proteins of NF4,5 and 6 are not cooperative, and we think that the one for the designed protein of NF3 is also not so cooperative. As described in the above response, we were not able to find flexible regions in these designed protein structures. Moreover, the broadened linewidths observed for many residues of NF4 in the previous Supplementary Fig. 4 were due to the setting of the ^1HN axis range. The replotted HSQC figures for all the designs with the same axis range do not show such broadened linewidths for all designs, including NF4 (new Supplementary Fig. 4). Therefore, we suppose that the non-cooperative behavior of thermal denaturation may originate from the design target topologies having locally interacting structural chunks, rather than the structure flexibility of the designs. NF3 and 4 contain strands sequentially aligned in a β -sheet (1234 and 4123, respectively), and NF5 and 6 contain one or two α -turn motifs, each of which interacts only with the β -sheet. We added the above discussion in the section “Experimental characterisation of novel four-stranded $\alpha\beta$ -folds” in the Results (Lines: 191-195).

Decision Letter, first revision:

Message: Our ref: NSMB-A45706A

30th Sep 2022

Dear Dr. Koga,

Thank you for submitting your revised manuscript "Exploration of novel $\alpha\beta$ -protein folds through de novo design" (NSMB-A45706A). I sincerely apologize for the unusual delay in responding, which was due to difficulty in reaching the original referees. Your manuscript has now been seen by two of the three original referees and their comments are below. The reviewers find that the paper has improved in revision, and therefore we'll be happy in principle to publish it in Nature Structural & Molecular Biology, pending minor revisions to satisfy the referees' final requests and to comply with our editorial and formatting guidelines.

To facilitate our work at this stage, we would appreciate if you could send us the main text as a word file. Please make sure to copy the NSMB account (cc'ed above).

Sincerely,
Sara

Sara Osman, Ph.D.
Associate Editor
Nature Structural & Molecular Biology

Reviewer #1 (Remarks to the Author):

All three reviewers thought this was an excellent manuscript in its original form. The reviewers suggested several minor changes. These changes have been incorporated into the revised manuscript. In my view the manuscript is now ready for publication.

Reviewer #3 (Remarks to the Author):

Points 1 and 2: OK

Point#3:

In looking at Supplementary Fig 9, it still seems that there are some discrepancies between the highly constrained structure and the TALOS+ order parameter.

For example, in NF2-02, the E1-E2 loop has $S_2 \sim 0.6$ (which indicates significant flexibility) for some residues but still has a highly ordered loop in the structure.

Conversely, in NF4-04, the E3-H3 loop looks disordered in the ensemble (consistent with the low number of NOE restraints) but the corresponding TALOS+ data suggests it is quite ordered.

In NF8-01, the H3-E4 loop looks well ordered from the ensemble but this is not reflected in the TALOS+ output, which suggests flexibility.

The most likely cause of these inconsistencies is that some loop regions have been over-constrained. This could be due to mis-assigned NOEs, increasing the number of restraints to flexible regions. (An alternative possibility is that some residues are mis-assigned, but that seems less likely).

Point#4:

The non-cooperative nature of the melts for NF3, 4, 5, and 6 is curious, since the HSQCs suggest well-folded proteins. The thermal denaturation profiles for NF4 and NF5 in particular look more like those of molten globules. The authors have attempted to address this in the text, but the explanation is not very convincing in the context of the tightly folded structures

that are presented. Could the lack of cooperativity in ~50% of the designs be part of the reason these folds have not been seen naturally? Can the authors comment on this further?

Regarding the revised HSQCs that are presented in the supplement, these are not as clear as what was presented in the first version. The spectra presented initially should be restored so the peaks can be seen clearly. The revised Fig4 has now decreased the intensity of the peaks so it is harder to see what is going on and it artificially makes the peaks look less broad. The difference in the NH axes is not so great between the spectra to account for the broadened signals in NF4 (e.g. Supp Fig1 and Supp Fig 4 were approximately the same range on the x-axis in the original manuscript, but the difference in peak widths is still evident).

These comments on the structure and stability data do not invalidate the work at all, but they should be addressed.

Author Rebuttal, first revision:

To Reviewer #1:

All three reviewers thought this was an excellent manuscript in its original form. The reviewers suggested several minor changes. These changes have been incorporated into the revised manuscript. In my view the manuscript is now ready for publication.

We sincerely appreciate your constructive and insightful comments and suggestions.

To Reviewer #3:

We sincerely appreciate your comments and suggestions. We revised our manuscript according to them. Point-by-point responses are described in the following.

Points 1 and 2: OK

Thank you.

Point#3:

In looking at Supplementary Fig 9, it still seems that there are some discrepancies between the highly constrained structure and the TALOS+ order parameter. For example, in NF2-02, the E1-E2 loop has S2 ~0.6 (which indicates significant flexibility) for some residues but still has a highly ordered loop in the structure. Conversely, in NF4-04, the E3-H3 loop looks disordered in the ensemble (consistent with the low number of NOE restraints) but the corresponding TALOS+ data suggests it is quite ordered. In NF8-01, the H3-E4 loop looks well ordered from the ensemble but this is not reflected in the TALOS+ output, which suggests flexibility. The most likely cause of these inconsistencies is that some loop regions have been over-constrained.

Thank you very much for the comments. It is difficult to discuss structural flexibility based solely on the TALOS+ prediction, and there is currently no established methodology in NMR structure determination to incorporate structural dynamics using NOE constraints. We also think that our

designed proteins consist of short loops, which would likely lead to converged structures even for the regions with the low order parameter, such as the E1-E2 loop in NF2-02 and the H3-E4 loop in NF8-01, which you pointed out. Note that the low structural convergence for the E3-H3 loop in

NF4-04 would be due to the lack of NOEs; the residues, 71, 74, and 75, in this region were not assigned.

This could be due to mis-assigned NOEs, increasing the number of restraints to flexible regions. (An alternative possibility is that some residues are mis-assigned, but that seems less likely).

While the possibility of mis-assigned NOEs or residues cannot be completely ruled out, it is unlikely because such cases typically lead to significant global conformational changes in structural determination. In these cases, the values such as R_p^{free} in the RDC validation tend to be low. However, since the RDC validation values are high (See Table 1), it is very difficult to consider the possibility of mis-assignments.

Finally, we realized that we need to describe how to understand regions with low order parameters. We excluded regions with the TALOS+ order parameter less than 0.8 in the RDC validation, so the accuracy of structural determination for regions with the low order parameters cannot be guaranteed. We added a sentence for this in the legend of Supplementary Fig. 11.

Point#4:

The non-cooperative nature of the melts for NF3, 4, 5, and 6 is curious, since the HSQCs suggest well-folded proteins. The thermal denaturation profiles for NF4 and NF5 in particular look more like those of molten globules. The authors have attempted to address this in the text, but the explanation is not very convincing in the context of the tightly folded structures that are presented. Could the lack of

cooperativity in ~50% of the designs be part of the reason these folds have not been seen naturally? Can the authors comment on this further?

We agree with you that our explanation on the non-cooperative nature of the melts for NF3, 4, 5, and 6 is not very convincing, so we deleted that description. As you suggested, the lack of cooperativity in ~50% of the designs could be part of the reason these folds have not been seen naturally. We added a sentence to describe this (Lines: 195-196).

Regarding the revised HSQCs that are presented in the supplement, these are not as clear as what was presented in the first version. The spectra presented initially should be restored so the peaks can be seen clearly. The revised Fig4 has now decreased the intensity of the peaks so it is harder to see what is going on and it artificially makes the peaks look less broad. The difference in the NH axes is not so great between the spectra to account for the broadened signals in NF4 (e.g. Supp Fig1 and Supp Fig 4 were approximately the same range on the x-axis in the original manuscript, but the difference in peak widths is still evident).

Following your suggestions, we revised the HSQC figures.

These comments on the structure and stability data do not invalidate the work at all, but they should be addressed.

Thank you, we have attempted to address them

Final Decision Letter:**Message** 30th May 2023

:

Dear Dr. Koga,

We are now happy to accept your revised paper "Exploration of novel $\alpha\beta$ -protein folds through de novo design" for publication as a Article in Nature Structural & Molecular Biology.

Your paper will be published online soon after we receive proof corrections and will appear in print in the next available issue. You can find out your date of online publication by contacting the production team shortly after sending your proof corrections. Content is published online weekly on Mondays and Thursdays, and the embargo is set at 16:00

London time (GMT)/11:00 am US Eastern time (EST) on the day of publication. Now is the time to inform your Public Relations or Press Office about your paper, as they might be interested in promoting its publication. This will allow them time to prepare an accurate and satisfactory press release. Include your manuscript tracking number (NSMB-A45706B) and our journal name, which they will need when they contact our press office.

About one week before your paper is published online, we shall be distributing a press release to news organizations worldwide, which may very well include details of your work. We are happy for your institution or funding agency to prepare its own press release, but it must mention the embargo date and Nature Structural & Molecular Biology. If you or your Press Office have any enquiries in the meantime, please contact press@nature.com.

Please note that *Nature Structural & Molecular Biology* is a Transformative Journal (TJ). Authors may publish their research with us through the traditional subscription access route or make their paper immediately open access through payment of an article-processing charge (APC). Authors will not be required to make a final decision about access to their article until it has been accepted. <https://www.springernature.com/gp/open-research/transformative-journals> Find out more about Transformative Journals

Authors may need to take specific actions to achieve [compliance](https://www.springernature.com/gp/open-research/funding/policy-compliance-faqs) with funder and institutional open access mandates. If your research is supported by a funder that requires immediate open access (e.g. according to [Plan S principles](https://www.springernature.com/gp/open-research/plan-s-compliance)) then you should select the gold OA route, and we will direct you to the compliant route where possible. For authors selecting the subscription publication route, the journal's standard licensing terms will need to be accepted, including [3](https://www.springernature.com/gp/open-research/policies/journal-

self-archiving policies. Those licensing terms will supersede any other terms that the author or any third party may assert apply to any version of the manuscript.

Sincerely,
Sara

Sara Osman, Ph.D.
Associate Editor
Nature Structural & Molecular Biology
